# Semi-gradient DICE for Offline Constrained Reinforcement Learning

## Abstract

Stationary Distribution Correction Estimation (DICE) addresses the mismatch between the stationary distribution induced by a policy and the target distribution required for reliable off-policy evaluation (OPE) and policy optimization. DICE-based offline constrained RL particularly benefits from the flexibility of DICE, as it simultaneously maximizes return while estimating costs in offline settings. However, we have observed that recent approaches designed to enhance the offline RL performance of the DICE framework inadvertently undermine its ability to perform OPE, making them unsuitable for constrained RL scenarios. In this paper, we identify the root cause of this limitation: their reliance on a semi-gradient optimization, which solves a fundamentally different optimization problem and results in failures in cost estimation. Building on these insights, we propose a novel method to enable OPE and constrained RL through semi-gradient DICE. Our method ensures accurate cost estimation and achieves state-of-the-art performance on the offline constrained RL benchmark, DSRL.

## 1. Introduction

Constrained reinforcement learning (RL) focuses on training agents to maximize return while adhering to predefined constraints, typically defined by a cost function. While conventional RL trains agents based on the interactions with the environment, such interactions in constrained environments may violate the constraints, which can be unsafe or prohibitively costly. To avoid the risk of constraint violations during online interactions, offline constrained RL has emerged as a practical solution. This approach relies on a fixed dataset of pre-collected experiences to train agents, eliminating the need for potentially unsafe online exploration during the training process.

Specifically, offline constrained RL aims to maximize the expected return of a policy while ensuring that predefined cost constraints are not violated, all within an offline setting. This requirement makes stationary distribution correction estimation (DICE) a promising framework, as it leverages stationary distribution to estimate and optimize the performance of a policy under both reward and cost functions simultaneously (Polosky et al., 2022; Lee et al., 2021b; Zhang et al., 2024). However, despite the theoretical soundness, prior research has largely been limited to finite domains or has struggled to achieve competitive performance when extending to continuous domains compared to algorithms from alternative frameworks.

Fortunately, recent empirical findings in offline RL (Sikchi et al., 2023; Mao et al., 2024b) indicate that incorporating a semi-gradient update into the DICE objective significantly improves training stability and achieves state-of-the-art RL performance in large and continuous domains. However, our analysis reveals that applying semi-gradient methods causes the DICE framework to lose its capability for off-policy evaluation (OPE), indicating that their effectiveness in offline RL does not generalize to constrained scenarios.

Although the semi-gradient DICE algorithms were adopted to stabilize conflicting gradients, inspired by the success of bootstrapped learning in deep RL algorithms, we discovered that they inherently solve a completely different optimization problem. This divergence leads to a solution with different characteristics. In this paper, we show that semi-gradient DICE algorithms are closely related to behavior-regularized offline RL and return a *policy correction* rather than the intended *stationary distribution correction*. This observation provides a partial explanation for the success behind the semi-gradient updates (Section 4).

Building on the analyses, we propose CORSDICE, an offline constrained RL algorithm that recovers a valid stationary distribution from the optimal policy correction of SemiDICE, thereby enabling OPE while maintaining the strong RL performance of semi-gradient DICE (Section 5). We provide empirical results that support the performance and validity of our method in offline RL, OPE, and offline constrained RL (Section 6).

---

[1]Anonymous Institution, Anonymous City, Anonymous Region, Anonymous Country. Correspondence to: Anonymous Author <anon.email@domain.com>.

Preliminary work. Under review by the International Conference on Machine Learning (ICML). Do not distribute.

## 2. Preliminary

**Markov Decision Process (MDP)** We model the environment as an infinite-horizon discounted Markov Decision Process (MDP) (Sutton & Barto, 2018), defined as $\mathcal{M} := \langle S, A, T, r, p_0, \gamma \rangle$, where $S$ is a set of states $s$, $A$ is a set of actions $a$, $T : S \times A \to \Delta(S)$ is a transition distribution, $r : S \times A \to \mathbb{R}$ is a reward function, $p_0 : \Delta(S)$ is a distribution over initial states $s_0$, and $\gamma \in [0, 1)$ is a discount factor. A policy $\pi : S \to \Delta(A)$ defines a distribution over actions that the agent selects given a state.

Given a policy $\pi$, its stationary distribution is defined as $d_\pi(s, a) := (1 - \gamma) \sum_{t=0}^{\infty} \gamma^t \Pr(s_t = s, a_t = a | \pi)$. This measures the probability of encountering a state-action pair $(s, a)$, when following policy $\pi$ in the discounted MDP $\mathcal{M}$. The stationary distributions satisfy the single-step transposed Bellman recurrence: $d_\pi(s, a) = (1 - \gamma) p_0(s) \pi(a \mid s) + \gamma \pi(a \mid s)(\mathcal{T}_* d)(s)$, where $(\mathcal{T}_* d)(s) := \sum_{\bar{s}, \bar{a}} T(s \mid \bar{s}, \bar{a}) d(\bar{s}, \bar{a})$. The expected discounted sum of rewards, or the value $\rho(\pi)$ of a policy $\pi$ can be evaluated using its corresponding stationary distribution $d_\pi$: $\rho(\pi) := (1 - \gamma) \mathbb{E}_\pi \left[ \sum_{t=0}^{\infty} \gamma^t r(s_t, a_t) \right] = \mathbb{E}_{(s,a) \sim d_\pi}[r(s, a)]$.

**OptiDICE** In Lee et al. (2021a), $f$-divergence regularization between the stationary distribution $d$ of the trained policy $\pi$ and the stationary distribution $d_D$ of the dataset policy $\pi_D$ is added to the objective to address the distribution shift issue in the offline RL.

$$\max_{d \geq 0} \sum_{s,a} d(s, a) r(s, a) - \alpha D_f(d || d_D) \quad (1a)$$

$$\text{s.t.} \sum_a d(s, a) = (1 - \gamma) p_0(s) + \gamma (\mathcal{T}_* d)(s), \; \forall s \quad (1b)$$

where $D_f(d || d_D) := \mathbb{E}_{(s,a) \sim d_D} \left[ f \left( \frac{d(s,a)}{d_D(s,a)} \right) \right]$ denotes $f$-divergence. The solution $d^*$ to this problem is the stationary distribution of the policy $\pi^*$ that maximizes the objective while adhering to Bellman flow constraints (1b).

OptiDICE (Lee et al., 2021a) is derived from the Lagrangian dual of (1) adopting the Lagrange multiplier $\nu(s)$ for (1b):

$$\min_\nu \max_{w \geq 0} \mathcal{L}(w, \nu) := (1 - \gamma) \mathbb{E}_{s_0 \sim p_0}[\nu(s_0)]$$
$$+ \mathbb{E}_{(s,a) \sim d_D} \left[ -\alpha f(w(s, a)) + w(s, a) e_\nu(s, a) \right] \quad (2)$$

where $d$ is replaced with the stationary distribution correction $w(s, a) := \frac{d(s,a)}{d_D(s,a)}$ to accommodate the offline dataset and $e_\nu(s, a) := r(s, a) + \gamma \mathbb{E}_{s'}[\nu(s')] - \nu(s)$ (derivation in Appendix A.1). Substituting $w$ with its closed-form solution $w_\nu^*(s, a) = \max \left( 0, (f')^{-1} \left( \frac{e_\nu(s,a)}{\alpha} \right) \right)$ gives:

$$\min_\nu (1 - \gamma) \mathbb{E}_{s_0 \sim p_0}[\nu(s_0)] + \alpha \mathbb{E}_{d_D} \left[ f_0^* \left( \frac{e_\nu(s,a)}{\alpha} \right) \right] \quad (3)$$

where $f_0^*(y) := \max_{x \geq 0} xy - f(x)$ is a convex conjugate of $f$ in $\mathbb{R}^+$. A notable benefit of optimizing the stationary dis-

tribution correction is its applicability to OPE for any reward function, expressed as $\hat{\rho}(\pi) = \mathbb{E}_{(s,a) \sim d_D}[w_{\nu^*}^*(s, a) r(s, a)]$.

After obtaining the solution $\nu^*$, we need to extract a policy $\pi^*$ that induces the stationary distribution $w_{\nu^*}^*(s, a) d_D(s, a)$. When we cannot do it analytically (e.g., continuous action space), assuming a parameterized policy $\pi_\theta$, we adopt weighted behavior cloning by minimizing:

$$\min_{\pi_\theta} -\mathbb{E}_{(s,a) \sim d_D}[w_{\nu^*}^*(s, a) \log \pi_\theta(a|s)] \quad (4)$$

Despite its elegant formulation, OptiDICE's performance in large and continuous domains falls short compared to other value-based offline RL algorithms (Mao et al., 2024b).

**Semi-gradient optimization** To improve the performance of OptiDICE, semi-gradient variants have been explored (Sikchi et al., 2023; Mao et al., 2024b). Since the residual $e_\nu(s, a)$ in (3) closely resembles the well-established Bellman error minimization framework (Sutton & Barto, 2018), incorporating semi-gradient optimization appears to be a natural extension, drawing inspiration from the success of fitted Q-iteration (Ernst et al., 2005).

Prior semi-gradient methods (Sikchi et al., 2023; Mao et al., 2024b) have involved three modifications: (1) partially or entirely omitting the gradient from the next state $\nu(s')$ in $e_\nu(s, a)$, (2) replacing the initial state distribution, $p_0(s)$ with the dataset distribution,[1] and (3) introducing a temperature hyperparameter $\beta$ to balance loss terms while removing $\alpha$. While (3) is widely adopted, it is primarily a design choice and not directly tied to the core semi-gradient optimization process.

To isolate and simplify the analysis of semi-gradient optimization, we adopt a semi-gradient DICE algorithm that incorporates only modifications (1) and (2), entirely omitting the gradient of $\nu(s')$ and replacing the initial state distribution with $d_D$. The resulting objectives are as follows:

$$\min_\nu \mathbb{E}_{(s,a) \sim d_D} \left[ \nu(s) + \alpha f_0^* \left( \frac{Q(s,a) - \nu(s)}{\alpha} \right) \right] \quad (5)$$

$$\min_Q \mathbb{E}_{(s,a,s') \sim d_D}[(r(s, a) + \gamma \nu(s') - Q(s, a)^2] \quad (6)$$

Note that we have adopted a new function approximator $Q$, which aligns the above problem with the semi-gradient optimization of (3) when $Q$ is an exact optimum. Moreover, this adoption removes the bias introduced by estimating the expectation within a convex function in (3) using finite samples, a limitation that renders OptiDICE biased in stochastic environments (Lee et al., 2021a; Kim et al., 2024a). We refer to this algorithm as **SemiDICE** throughout the paper (details in Appendix B).

---

[1]Interestingly, when the dataset distribution satisfies Bellman flow constraint (i.e., corresponds to $d_D$) modification (2) theoretically has no effect under a semi-gradient update that entirely omits the gradient of $\nu(s')$ (see Appendix B).

Based on the semi-gradient optimization of DICE, Sikchi et al. (2023); Mao et al. (2024b) achieved state-of-the-art performance at the time of their publication. Similarly, our empirical results confirm that SemiDICE demonstrates strong offline RL performance, exhibiting behavior consistent with those previous studies.

## 3. Constrained RL with DICE

In this section, we revisit the extension of the OptiDICE algorithm to constrained RL problems and explore the feasibility of extending SemiDICE using a similar approach.

**Constrained RL**   The constrained RL (Altman, 1999) aims to obtain a policy that maximizes an expected return while satisfying cost constraints defined by a cost function $c : S \times A \to \mathbb{R}$ and a cost threshold $C_{\lim} \in \mathbb{R}$. The objective can be formulated as:

$$\max_{\pi} \mathbb{E}_{\pi} \Big[ \sum_{t=0}^{\infty} \gamma^t r(s_t, a_t) \Big] \text{ s.t. } \mathbb{E}_{\pi} \Big[ \sum_{t=0}^{\infty} \gamma^t c(s_t, a_t) \Big] \le C_{\lim} \quad (7)$$

**COptiDICE**   The convenience of OPE using stationary distributions naturally extends to estimating the discounted sum of costs. In Lee et al. (2021b), the constrained extension of OptiDICE solves Eq. (1) with the inclusion of additional constraints:

$$\sum_{s,a} d(s,a) c(s,a) \le (1 - \gamma) C_{\lim} =: \tilde{C}_{\lim}. \quad (8)$$

To satisfy the constraint, COptiDICE formulates the Lagrangian dual by adopting Lagrangian multiplier $\lambda$ for the cost constraint Eq. (8):

$$\min_{\nu, \lambda \ge 0} \max_{w \ge 0} \mathcal{L}(w, \nu) - \lambda \mathbb{E}_{d_D}[w(s,a)c(s,a) - \tilde{C}_{\lim}])$$

where $d$ is replaced with $w$ as in Eq. (2). Similar to OptiDICE, we can solve for the closed-form solution of $w$ and substitute it in to get training objective of $\nu$:

$$\min_{\nu} (1 - \gamma) \mathbb{E}_{s_0 \sim p_0}[\nu(s_0)] + \alpha \mathbb{E}_{d_D} \Big[ f_0^* \big( \tfrac{e_{\nu,\lambda}(s,a)}{\alpha} \big) \Big]$$

$$\min_{\lambda \ge 0} \lambda \Big( \tilde{C}_{\lim} - \mathbb{E}_{(s,a) \sim d_D}[w_{\nu,\lambda}^*(s,a)c(s,a)] \Big)$$

$$w_{\nu,\lambda}^*(s,a) = \max \big( 0, (f')^{-1} \big( \tfrac{e_{\nu,\lambda}(s,a)}{\alpha} \big) \big)$$

where $e_{\nu,\lambda}(s,a) := r(s,a) - \lambda c(s,a) + \gamma \mathbb{E}_{s'}[\nu(s')] - \nu(s)$ (derivation in Appendix A.2). Looking at how $e_{\nu,\lambda}$ differs from previous $e_{\nu}$, we can interpret this algorithm as solving OptiDICE with a penalized reward function, $r(s,a) - \lambda c(s,a)$, where $\lambda$ is adjusted based on the cost constraint: it increases when the constraint is violated and decreases otherwise.

**Constrained SemiDICE**   As COptiDICE naturally extends OptiDICE to constrained RL, it initially appears feasible to extend SemiDICE in a similar manner to formulate a constrained SemiDICE, potentially enhancing performance in constrained RL problems. Naively applying the modifications introduced in SemiDICE to COptiDICE results in the following:

$$\min_{\nu} \mathbb{E}_{d_D} \Big[ \nu(s) + \alpha f_0^* \big( \tfrac{Q(s,a) - \nu(s)}{\alpha} \big) \Big] \quad (9)$$

$$\min_{Q} \mathbb{E}_{d_D} \Big[ (r(s,a) - \lambda c(s,a) + \gamma \nu(s') - Q(s,a))^2 \Big] \quad (10)$$

$$\min_{\lambda \ge 0} \lambda \Big( \tilde{C}_{\lim} - \mathbb{E}_{(s,a) \sim d_D}[w_{\nu,\lambda}^*(s,a)c(s,a)] \Big) \quad (11)$$

where $w_{\nu,\lambda}^*(s,a) = \max \big( 0, (f')^{-1} \big( \tfrac{Q(s,a) - \nu(s)}{\alpha} \big) \big)$.

However, as will be described in Section 4 and Table 1, this naive constrained SemiDICE completely fails to satisfy the cost constraint due to its inability to perform OPE correctly: SemiDICE estimates policy corrections rather than stationary distribution corrections, making $\mathbb{E}_{d_D}[w_{\nu,\lambda}^*(s,a)c(s,a)]$ no longer a valid cost estimate. In the following sections, we analyze the root cause of this inability to conduct OPE and propose a method to address this issue.

## 4. Demystifying SemiDICE

In this section, we discuss various characteristics of the SemiDICE algorithm that deepen our understanding of semi-gradient optimization within the DICE framework.

**Solution of SemiDICE**   We show that the correction $w(s,a)$ obtained by solving SemiDICE is not a stationary distribution correction, but rather a policy correction.

**Proposition 4.1** (Solution characteristics of SemiDICE). *The correction $w^*(s,a)$ obtained by the optimal $\nu^* = \arg\min_{\nu} \mathbb{E}_{d_D} \big[ \nu(s) + \alpha f_0^* \big( \tfrac{Q(s,a) - \nu(s)}{\alpha} \big) \big]$,*

$$w^*(s,a) = \max \big( 0, (f')^{-1} \big( \tfrac{Q(s,a) - \nu^*(s)}{\alpha} \big) \big), \quad (12)$$

*violates the Bellman flow constraint* (1b) *but satisfies the following conditions for $w^*(s,a)$ to act as a policy correction (Proof in Appendix B.1):*

$$\sum_{a} w^*(s,a) \pi_D(a|s) = 1, \ w^*(s,a) \ge 0, \ \forall s, a. \quad (13)$$

Proposition 4.1 explains the failure of the naive constrained SemiDICE as $w(s,a)$ no longer converges to a stationary distribution correction under semi-gradient optimization, and resulting policy correction is incapable of performing OPE. The semi-gradient update has caused $\nu$ to lose its role as Lagrangian multiplier that ensures the satisfaction of Bellman flow constraints (1b). Similarly, other DICE algorithms employing semi-gradient optimization also fail to

converge to stationary distribution corrections, instead converging to constant multiples of policy corrections (Sikchi et al., 2023) or somewhere in-between two corrections (Mao et al., 2024b) (Appendices B.1 and B.2). For policy extraction, since weighted behavior cloning (Eq. (4)) minimizes $\text{KL}\big(\frac{w(s,a)d_D(s,a)}{\sum_a w(s,a)d_D(s,a)} \| \pi_\theta(a|s)\big)$, it remains effective regardless of the solution characteristics and has been successfully applied in previous studies. As $w^*(s,a)$ computed by SemiDICE (Eq. (12)) represents the policy correction, we will denote it as $w(a|s)$ in later sections.

**Connections to behavior-regularized offline RL** While we have identified that SemiDICE results in a policy correction, it remains unclear what specific problem it addresses. We now provide an explanation of the problem that SemiDICE solves. Building upon the findings of prior works (Xu et al., 2022; Sikchi et al., 2023), we demonstrate that SemiDICE, SQL (Xu et al., 2022) and XQL (Garg et al., 2023) solve behavior-regularized MDP introduced in offline RL (Xu et al., 2022) with different approximations.

We begin with the behavior-regularized MDP (Xu et al., 2022; Geist et al., 2019), where reward is penalized by the $f$-divergence between the dataset policy $\pi_D$ and policy $\pi$. However, we reverse the divergence to get the corresponding $f$-divergence regularization between $\pi$ and $\pi_D$:

$$\max_\pi \mathbb{E}_\pi \Big[ \sum_{t=0}^\infty \gamma^t \big( r(s_t, a_t) - \alpha \frac{\pi_D(a_t|s_t)}{\pi(a_t|s_t)} f\big( \frac{\pi(a_t|s_t)}{\pi_D(a_t|s_t)} \big) \big) \Big].$$

Note that this still qualifies as a behavior-regularized MDP, as $xf(1/x)$ satisfies the necessary conditions for an $f$-divergence. The policy evaluation operator of the regularized MDP is given by,

$$(\mathcal{T}_f^\pi Q)(s,a) := r(s,a) + \gamma \mathbb{E}_{s' \sim T(\cdot|s,a)}[V(s')]$$

where $V(s) := \mathbb{E}_{a \sim \pi} \Big[ Q(s,a) - \alpha \frac{\pi_D(a|s)}{\pi(a|s)} f\big( \frac{\pi(a|s)}{\pi_D(a|s)} \big) \Big]$.

**Proposition 4.2.** *In the behavior-regularized MDP, the optimal value functions $V^*(s), Q^*(s,a)$ and the optimal policy correction $\frac{\pi^*(a|s)}{\pi_D(a|s)}$ of the regularized MDP are given by (Proof in Appendix C.2):*

$$U^*(s) = \underset{U(s)}{\arg \min} \ U(s) + \mathbb{E}_{a \sim \pi_D} \big[ \alpha f^* \big( \frac{Q^*(s,a) - U(s)}{\alpha} \big) \big]$$

$$V^*(s) = U^*(s) + \mathbb{E}_{a \sim \pi_D} \big[ \alpha f^* \big( \frac{Q^*(s,a) - U^*(s)}{\alpha} \big) \big]$$

$$Q^*(s,a) = r(s,a) + \gamma \mathbb{E}_{s' \sim T}[V^*(s')]$$

$$\frac{\pi^*(a|s)}{\pi_D(a|s)} = \max \big( 0, (f')^{-1} \big( \frac{Q^*(s,a) - U^*(s)}{\alpha} \big) \big)$$

From this, we can observe that SemiDICE is equivalent to a behavior-regularized RL that approximates $V^*(s)$ of the regularized MDP with $U^*(s)$, while SQL approximates $V^*(s)$ with $U^*(s) + \alpha$ when $f(x) = x^2 - x$, and XQL is a special case without any approximation in $V^*(s)$ when $f(x) = x \log x$ (details in Appendix C).

**Advantage of semi-gradient update** With the characteristics of SemiDICE clarified, we now offer an additional explanation for why SemiDICE generally outperforms OptiDICE in large, continuous domains. Beyond the stabilization of updates, which previous studies have identified as a key factor for improvement (Sikchi et al., 2023; Mao et al., 2024b), our analysis reveals an alternative perspective.

A key issue with OptiDICE arises because, as the policy is optimized, the support of both the policy and the states it visits tends to shrink. Depending on $f$-divergence, there often exist a state $s$ such that $d_{\pi^*}(s,a) = 0$ for all $a$, which implies that $\pi^*(a|s)$ is undefined for that state, even if $s$ appears in the dataset. This sparsity problem gets worse as the state space gets larger. On the other hand, even when using the same $f$-divergence that can induce sparse solutions, semi-gradient-based DICE methods yield a sparse optimal **policy** (similar to Xu et al. 2022) rather than a sparse optimal **state-action stationary distribution**, allowing them to avoid issues caused by state distribution sparsity.

In response to such worst-case scenarios where $d_{\pi^*}(s,a) = w^*(s,a) = 0$, previously proposed DICE algorithms either resort to a uniform random policy (for finite action space; Zhan et al. 2022; Ozdaglar et al. 2023; Zhang et al. 2024) or refrain from updating the policy for the corresponding state $s$ (see Eq. (4)), resulting in data inefficiency. In contrast, SemiDICE, which is guaranteed to output policy corrections, eliminates this issue and avoids the data inefficiency caused by state sparsity.

**Corollary 4.3** (SemiDICE avoiding sparsity problem). *Let $w^*$ be the correction (Eq. (12)) optimized by running SemiDICE. There is no state $s$ where $w^*(s,a) = 0 \ \forall a$. (Proof in Appendix C.3)*

## 5. CORSDICE: semi-gradient DICE for offline constrained RL

In this section, we fix the off-policy cost evaluation in Eq. (11) of constrained SemiDICE, building on the finding from the previous section that *the solution of SemiDICE is a policy correction*. Let the policy correction found by SemiDICE (Eq. (12)) be defined as $w(a|s) = \frac{\pi_w(a|s)}{\pi_D(a|s)}$, where $\pi_w$ is the policy we get when we extract policy from the solution of SemiDICE. To enable OPE with respect to $w(a|s)$, we need to compute the state stationary distribution correction $w(s) := \frac{d_w(s)}{d_D(s)} = \frac{w(s,a)}{w(a|s)}$, where $d_w(s)$ is the state stationary distribution induced by $\pi_w$. If we can successfully compute $w(s)$, off-policy cost evaluation becomes possible with $w(s)w(a|s)$, allowing us to optimize $\lambda$ using the following:

$$\min_{\lambda \geq 0} \ \lambda \big( \tilde{C}_{\text{lim}} - \underbrace{\mathbb{E}_{(s,a) \sim d_D}[w(s)w(a|s)c(s,a)]}_{=\mathbb{E}_{(s,a) \sim d^\pi}[c(s,a)]} \big). \quad (14)$$

## 5.1. State stationary distribution extraction

To compute the state stationary distribution correction $w(s)$ given the policy correction $w(a|s)$ obtained from SemiDICE, we introduce the following novel optimization problem:

$$\max_{w(s) \geq 0} \ -\sum_s d_D(s) f(w(s)) \tag{15a}$$

$$\text{s.t. } w(s) d_D(s) = (1 - \gamma) p_0(s) + \gamma (\mathcal{T}_* d_w)(s) \ \forall s \tag{15b}$$

where $(\mathcal{T}_* d_w)(s) := \sum_{\bar{s}, \bar{a}} T(s \mid \bar{s}, \bar{a}) w(\bar{s}) w(\bar{a}|\bar{s}) d_D(\bar{s}, \bar{a})$.

Note that the $|S|$ constraints (15b) in this problem uniquely determine $w(s)$, regardless of the objective, making the problem over-constrained. However, incorporating the $f$-divergence minimization objective between $d_w(s)$ and $d_D(s)$ (15a) adds convexity to the optimization, facilitating efficient sample-based optimization, similar to the dual of Q-LP and related algorithms (Nachum & Dai, 2020).

The Lagrangian dual of the problem, with Lagrange multipliers $\mu(s)$ for the constraint (15b), is given as (Full derivation in Appendix D):

$$\max_{w(s) \geq 0} \min_{\mu(s)} (1 - \gamma) \mathbb{E}_{s_0 \sim p_0} [\mu(s_0)] \tag{16}$$

$$+ \mathbb{E}_{(s,a) \sim d_D} [w(s) w(a|s) e_\mu(s, a) - f(w(s))],$$

where $e_\mu(s, a) = \gamma \sum_{s'} T(s'|s, a) \mu(s') - \mu(s)$. We obtain a closed-form solution of $w^*(s)$ by reversing the optimization order based on strong duality of Eq. (15):

$$w^*(s) = \max(0, (f')^{-1}(\mathbb{E}_{a \sim \pi_D}[w(a|s) e_\mu(s, a)])). \tag{17}$$

Substituting $w^*(s)$ into Eq. (16) results in the following:

$$\min_{\mu} \ \mathcal{L}_{\text{ext}}(\mu) := (1 - \gamma) \mathbb{E}_{s_0 \sim p_0} [\mu(s_0)] \tag{18}$$

$$+ \mathbb{E}_{s \sim d_D} [f_0^*(\mathbb{E}_{a \sim \pi_D}[w(a|s) e_\mu(s, a)])].$$

However, sample-based optimization of $\mathcal{L}_{\text{ext}}$ poses a challenge due to the presence of expectations over the transition $T$ and the dataset policy $\pi_D$ within the convex function $f_0^*(x)$. Unlike previous algorithms, which are biased only in the presence of stochastic transitions (Lee et al., 2021a;b), the common stochasticity in the dataset policy introduces significant bias when these expectations are estimated using a single sample.

**Bias reduction for sample-based optimization** To address the issue, we propose a simple bias reduction technique by incorporating an additional function approximator, $A(s)$, to estimate the expectation inside $f_0^*(\cdot)$. We then decompose the $\mu$ optimization of (18) into the following optimizations on $A$ and $\mu$, which share the same optimal

---

**Algorithm 1** CORSDICE

**Input**: Dataset $D$, Initial state dataset $p_0, \alpha$
**Parameter**: $\psi, \phi, \xi, \zeta, \theta, \lambda$
**Output**: A policy $\pi_\theta$

    Let $t = 0$
    Initialize parameters
    **for** $t = 1, 2, \ldots, N$ **do**
        Sample from $(s, a, r, c, s') \sim D$, $s_0 \sim \mu_0$
        # SemiDICE for penalized reward $r(s, a) - \lambda c(s, a)$
        Update $\nu_\psi, Q_\phi$ using (9), (10) given $\lambda$
        # Estimating state distribution correction $w(s)$
        $w(a|s) := \max\left(0, (f')^{-1}\left(\frac{Q_\phi(s,a) - \nu_\psi(s)}{\alpha}\right)\right)$
        Update $A_\xi, \mu_\zeta$ using (19a), (19b) given $w(a|s)$
        $w(s) := \max\left(0, (f')^{-1}(A_\xi(s))\right)$
        # Updating the cost Lagrange multiplier
        Update $\lambda$ using (14) given $w(s), w(a|s)$
        # Policy extraction step
        Update $\pi_\theta$ using (4) given $w(a|s)$
    **end for**

---

solution of $\mu$:

$$\min_A \mathbb{E}_{(s,a,s') \sim d_D} \left[ \left( A(s) - w(a|s) \hat{e}_\mu(s, s') \right)^2 \right] \tag{19a}$$

$$\min_\mu \tilde{\mathcal{L}}_{\text{ext}}(\mu) := (1 - \gamma) \mathbb{E}_{s_0 \sim p_0} [\mu(s_0)] \tag{19b}$$

$$+ \mathbb{E}_{(s,a,s') \sim d_D} [(f_0^*)'(A(s)) w(a|s) \hat{e}_\mu(s, s')],$$

where $\hat{e}_\mu(s, s') = \gamma \mu(s') - \mu(s)$.

**Proposition 5.1.** *Minimization of the objectives in* (19) *results in the same optimal* $\mu^*$ *as in* (18)*. (Proof in Appendix D)*

In $\tilde{\mathcal{L}}_{\text{ext}}(\mu)$, the expectations are moved outside the nonlinear function, enabling an unbiased sample-based estimation. In practice, $A(s)$ is parameterized and estimated using a neural network. While function approximation errors may introduce some additional bias, our empirical observations suggest that this bias is significantly smaller than the bias introduced by relying on the naive single-sample estimator for evaluating $\mathcal{L}_{\text{ext}}(\mu)$. After the optimizations on (19) are complete, the state stationary distribution correction $w(s)$ is naturally computed as $w(s) = \max(0, (f')^{-1}(A^*(s)))$.

## 5.2. CORSDICE: putting things together

By optimizing Eq. (19a-19b), we obtain the stationary distribution correction $w(s) w(a|s)$ corresponds to the policy optimized by SemiDICE, enabling accurate off-policy cost evaluation for Eq. (14). We present ***Constrained Offline RL via Semi-gradient stationary DIstribution Correction Estimation*** (CORSDICE), which alternates SemiDICE and off-policy cost evaluation through stationary distribution extraction in each iteration. SemiDICE optimizes the policy

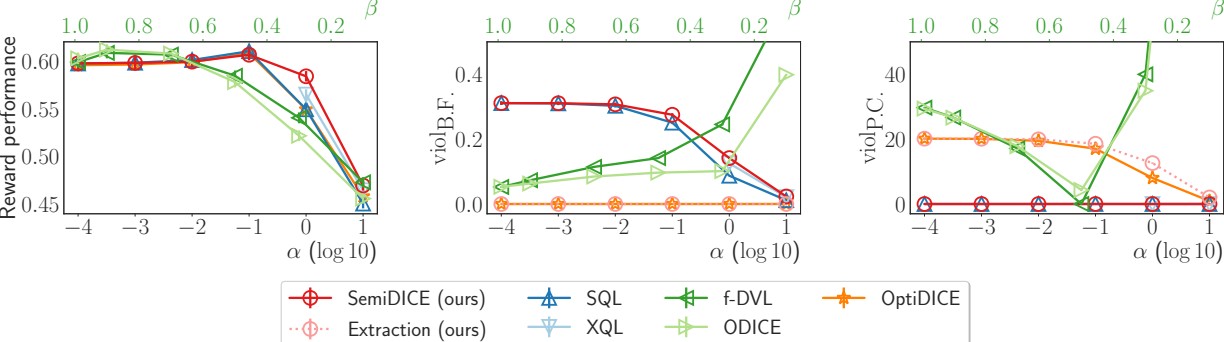

*Figure 1.* Return (**Left**), Bellman flow constraint violation (**Middle**), and policy correction constraint violation (**Right**). Results are averaged over 300 runs. The hyperparameters $\alpha$ (SemiDICE, SQL, XQL, OptiDICE) and $\beta$ ($f$-DVL and ODICE) determine the degree of $f$-divergence regularization, where the regularization becomes stronger as $\alpha$ increases and $\beta$ decreases. The performance of **XQL** for small $\alpha$ values is omitted due to numerical stability issues.

correction for the penalized reward signal $r(s,a) - \lambda c(s,a)$, and $\lambda$ is updated based on the off-policy cost evaluation. Algorithm 1 outlines the complete set of learning objectives for CORSDICE.

# 6. Experiment

We conduct three experiments to empirically validate our findings and showcase the performance of CORSDICE.

## 6.1. Examining algorithm characteristics

We evaluate offline RL algorithms on a randomly generated finite MDP experiment (Laroche et al., 2019; Lee et al., 2020) (details in Appendix F.1). We compare four DICE-based RL algorithms (**OptiDICE**, **SemiDICE**, **f-DVL**, **ODICE**), two behvaior-regularized RL algorithms (**SQL**, **XQL**), and **Extraction**, an application of the state stationary distribution extraction method to **SemiDICE**.

We analyze the algorithms across three aspects, visualized in Figure 1: **left**, policy performance $\rho(\pi)$; **middle**, violation of the Bellman flow constraint ($\text{viol}_{\text{B.F.}}$); and **right**, violation of the policy correction constraint ($\text{viol}_{\text{P.C.}}$). Violations are quantified using the $L_1$-norm:

$$\text{viol}_{\text{B.F.}} = \sum_s |(1-\gamma)p_0(s) + \gamma(\mathcal{T}_* d_w)(s) - (\mathcal{B}_* d_w)(s)|,$$

$$\text{viol}_{\text{P.C.}} = \sum_s \left| \sum_a w(s,a)\pi_D(a|s) - 1 \right|.$$

**Reward performance** The reward performance of all offline RL algorithms initially improves as $\alpha$ increases (as $\beta$ decreases), but then declines as $\alpha$ becomes large (when $\beta$ is small) due to stronger conservatism shifting the policy toward suboptimal datasets. In tabular domains, when hyperparameters are properly tuned, performance differences among full-gradient (OptiDICE), semi-gradient (SemiDICE,

*Table 1.* RMSE of OPE for SemiDICE policies trained on a subset of the D4RL (Fu et al., 2020) dataset (more results in Appendix H.2).

| ALGORITHM | HOPPER | HALFCHEETAH | WALKER2D |
|---|---|---|---|
| SEMIDICE | 90.62 | 87.58 | 111.63 |
| EXTRACTION (OURS) | **20.70** | **26.44** | **9.20** |
| DUALDICE | 58.08 | 162.81 | 20.53 |
| IHOPE | 78.61 | 57.92 | 90.01 |

f-DVL), and orthogonal-gradient (ODICE) are not significant. However, the benefits of semi-gradient optimization over full-gradient optimization become more evident in large and continuous domain experiments leading to significant difference in returns (see Appendix F.2).

**Solution characteristics** Our primary focus in this experiment is to assess whether semi-gradient DICE methods produce policy corrections rather than stationary distribution corrections (Proposition 4.1). Figure 1-Middle shows that only **OptiDICE** and **Extraction** satisfy the Bellman-flow constraint (zero violation). In contrast, **SemiDICE**, **SQL**, **XQL**, and **f-DVL** ($\beta = 0.5$) yield policy corrections instead of stationary distribution corrections (Figure 1-Right). These empirical results align with our theoretical findings: (1) SemiDICE outputs policy corrections, (2) SemiDICE is closely related to behavior-regularized RL algorithms, and (3) Extraction produces stationary distribution corrections from the policy corrections provided by SemiDICE.

## 6.2. Experiments on Off-Policy Evaluation

In this experiment, we estimate the returns of offline RL policies pre-trained with SemiDICE on three D4RL (Fu et al., 2020) benchmarks. Table 1 presents RMSE between the estimated and the average discounted returns.

We compare four algorithms: **SemiDICE**, which directly

*Table 2.* Normalized DSRL (Liu et al., 2024) benchmark results, averaged over 5 seeds and 20 episodes. **Gray**: Unsafe agents, **Bold**: Safe agents with normalized costs below 1.0, **Blue**: Safe agents achieving the highest normalized return.

| TASK | BC-ALL | | BC-SAFE | | BCQ-LAG | | BEAR-LAG | | CPQ | | COptiDICE | | CORSDICE (OURS) | |
|---|---|---|---|---|---|---|---|---|---|---|---|---|---|---|
| | REWARD↑ | COST↓ | REWARD↑ | COST↓ | REWARD↑ | COST↓ | REWARD↑ | COST↓ | REWARD↑ | COST↓ | REWARD↑ | COST↓ | REWARD↑ | COST↓ |
| POINTBUTTON1 | 0.14 | 1.01 | 0.06 | 0.60 | 0.38 | 2.69 | 0.60 | 3.47 | 0.70 | 4.07 | 0.15 | 1.00 | 0.22 | 0.94 |
| POINTBUTTON2 | 0.26 | 1.62 | 0.16 | 1.02 | 0.45 | 2.78 | 0.65 | 3.63 | 0.64 | 3.30 | 0.26 | 1.61 | 0.13 | 0.98 |
| POINTCIRCLE1 | 0.78 | 4.80 | 0.41 | 0.20 | 0.83 | 4.65 | 0.34 | 2.31 | 0.54 | 0.29 | 0.80 | 4.00 | 0.43 | 0.93 |
| POINTCIRCLE2 | 0.67 | 4.89 | 0.47 | 0.96 | 0.65 | 3.77 | 0.26 | 3.84 | 0.32 | 1.18 | 0.64 | 4.18 | 0.49 | 0.75 |
| POINTGOAL1 | 0.64 | 0.93 | 0.42 | 0.35 | 0.72 | 1.02 | 0.77 | 1.18 | 0.44 | 0.62 | 0.63 | 0.96 | 0.75 | 0.89 |
| POINTGOAL2 | 0.53 | 2.00 | 0.27 | 0.76 | 0.74 | 3.72 | 0.84 | 3.94 | 0.50 | 1.26 | 0.55 | 2.08 | 0.37 | 0.85 |
| POINTPUSH1 | 0.23 | 0.88 | 0.16 | 0.52 | 0.36 | 1.08 | 0.44 | 1.01 | 0.26 | 1.27 | 0.24 | 0.74 | 0.27 | 0.81 |
| POINTPUSH2 | 0.14 | 1.21 | 0.12 | 0.59 | 0.25 | 1.51 | 0.27 | 1.81 | 0.14 | 1.55 | 0.15 | 0.86 | 0.18 | 0.07 |
| CARBUTTON1 | 0.16 | 1.73 | 0.05 | 0.50 | 0.44 | 7.50 | 0.53 | 7.49 | 0.53 | 8.26 | 0.00 | 1.40 | 0.09 | 0.48 |
| CARBUTTON2 | -0.13 | 1.78 | 0.03 | 0.67 | 0.53 | 6.12 | 0.60 | 6.24 | 0.61 | 5.03 | -0.04 | 1.23 | 0.06 | 0.71 |
| CARCIRCLE1 | 0.72 | 5.32 | 0.30 | 1.32 | 0.76 | 4.95 | 0.81 | 6.78 | 0.03 | 2.41 | 0.71 | 4.91 | -0.05 | 0.64 |
| CARCIRCLE2 | 0.69 | 6.42 | 0.40 | 2.19 | 0.69 | 6.18 | 0.83 | 10.45 | 0.52 | 0.41 | 0.68 | 6.00 | 0.33 | 0.78 |
| CARGOAL1 | 0.40 | 0.54 | 0.29 | 0.39 | 0.50 | 0.95 | 0.71 | 1.29 | 0.81 | 0.94 | 0.51 | 0.82 | 0.53 | 0.79 |
| CARGOAL2 | 0.28 | 1.06 | 0.16 | 0.49 | 0.69 | 3.51 | 0.83 | 3.74 | 0.88 | 4.26 | 0.33 | 1.24 | 0.39 | 0.99 |
| CARPUSH1 | 0.22 | 0.56 | 0.18 | 0.46 | 0.36 | 0.73 | 0.43 | 0.83 | 0.15 | 1.33 | 0.22 | 0.56 | 0.22 | 0.71 |
| CARPUSH2 | 0.12 | 1.49 | 0.05 | 0.40 | 0.38 | 2.68 | 0.35 | 2.78 | 0.29 | 3.62 | 0.13 | 1.15 | 0.15 | 0.91 |
| SWIMMERVEL | 0.47 | 1.05 | 0.47 | 0.31 | 0.28 | 2.35 | 0.17 | 0.84 | 0.59 | 3.18 | 0.56 | 0.64 | 0.09 | 0.38 |
| HOPPERVEL | 0.85 | 3.78 | 0.61 | 2.11 | 0.46 | 2.18 | -0.01 | 0.0 | 0.46 | 3.08 | 0.86 | 1.60 | 0.80 | 0.41 |
| HALFCHEETAHVEL | 0.89 | 2.57 | 0.88 | 0.13 | 0.90 | 1.16 | 0.03 | 0.01 | 0.32 | 0.95 | 0.96 | 0.24 | 0.95 | 0.30 |
| WALKER2DVEL | 0.81 | 1.19 | 0.79 | 0.00 | 0.64 | 2.53 | 0.01 | 0.00 | 0.00 | 0.02 | 0.80 | 0.87 | 0.80 | 0.02 |
| ANTVEL | 0.98 | 4.73 | 0.97 | 0.36 | 0.66 | 4.03 | 0.32 | 0.12 | -0.14 | 0.04 | 1.01 | 1.24 | 0.98 | 0.23 |
| **SAFETYGYM AVERAGE** | 0.47 | 2.36 | 0.35 | 0.68 | 0.56 | 3.15 | 0.47 | 2.94 | 0.41 | 2.19 | 0.48 | 1.78 | 0.40 | 0.71 |
| BALLRUN | 0.43 | 1.10 | 0.25 | 1.15 | 0.96 | 2.49 | 0.02 | 1.56 | 0.64 | 2.70 | -0.01 | 0.00 | 0.25 | 0.99 |
| CARRUN | 0.97 | 0.15 | 0.97 | 0.12 | 0.96 | 0.32 | -0.54 | 0.37 | 0.89 | 0.41 | 0.30 | 0.01 | 0.97 | 0.55 |
| DRONERUN | 0.56 | 1.73 | 0.43 | 1.14 | 0.58 | 1.96 | -0.18 | 5.40 | 0.40 | 1.40 | 0.59 | 1.42 | 0.48 | 0.97 |
| ANTRUN | 0.73 | 1.55 | 0.69 | 0.95 | 0.63 | 0.92 | 0.27 | 0.25 | 0.19 | 0.43 | 0.73 | 1.35 | 0.66 | 0.51 |
| BALLCIRCLE | 0.72 | 1.13 | 0.40 | 0.55 | 0.87 | 1.52 | 0.31 | 1.54 | 0.74 | 0.75 | 0.78 | 1.18 | 0.56 | 0.64 |
| CARCIRCLE | 0.72 | 1.11 | 0.18 | 1.11 | 0.65 | 2.48 | 0.15 | 2.5 | 0.70 | 0.66 | 0.76 | 1.28 | 0.34 | 0.64 |
| DRONECIRCLE | 0.68 | 1.17 | 0.55 | 0.42 | 0.50 | 0.24 | -0.11 | 0.35 | -0.11 | 1.31 | 0.84 | 1.16 | 0.55 | 0.77 |
| ANTCIRCLE | 0.71 | 2.83 | 0.48 | 1.26 | 0.84 | 4.29 | 0.22 | 0.52 | 0.00 | 0.00 | 0.74 | 5.39 | 0.53 | 0.88 |
| **BULLETGYM AVERAGE** | 0.69 | 1.35 | 0.49 | 0.84 | 0.75 | 1.78 | 0.02 | 1.56 | 0.43 | 0.96 | 0.59 | 1.47 | 0.54 | 0.77 |
| EASYSPARSE | 0.27 | 0.32 | 0.32 | 0.06 | 1.37 | 3.10 | -0.03 | 0.05 | -0.23 | 0.17 | 0.91 | 2.64 | 0.54 | 0.85 |
| EASYMEAN | 0.51 | 1.45 | 0.25 | 0.00 | 1.31 | 2.67 | -0.03 | 0.07 | -0.06 | 0.02 | 0.75 | 2.67 | 0.49 | 0.91 |
| EASYDENSE | 0.64 | 2.21 | 0.22 | 0.01 | 1.02 | 1.99 | 0.09 | 0.46 | -0.06 | 0.02 | 0.70 | 1.13 | 0.52 | 0.94 |
| MEDIUMSPARSE | 0.81 | 1.15 | 0.74 | 0.14 | 0.77 | 0.72 | -0.03 | 0.02 | -0.08 | 0.01 | 0.83 | 1.51 | 0.99 | 0.94 |
| MEDIUMMEAN | 0.77 | 1.37 | 0.72 | 0.25 | 2.03 | 2.60 | -0.02 | 0.03 | -0.08 | 0.02 | 0.92 | 1.89 | 0.96 | 0.76 |
| MEDIUMDENSE | 0.81 | 1.26 | 0.82 | 0.82 | 2.20 | 2.79 | 0.06 | 0.16 | -0.07 | 0.00 | 0.73 | 0.89 | 0.98 | 0.83 |
| HARDSPARSE | 0.46 | 2.07 | 0.37 | 0.19 | 1.15 | 2.78 | 0.01 | 0.28 | -0.04 | 0.01 | 0.56 | 1.64 | 0.41 | 0.84 |
| HARDMEAN | 0.36 | 1.14 | 0.32 | 0.08 | 0.94 | 2.18 | 0.00 | 0.11 | -0.05 | 0.01 | 0.64 | 1.14 | 0.42 | 0.83 |
| HARDDENSE | 0.40 | 1.70 | 0.29 | 0.10 | 1.19 | 3.00 | 0.00 | 0.05 | -0.04 | 0.00 | 0.51 | 0.72 | 0.27 | 0.59 |
| **METADRIVE AVERAGE** | 0.56 | 1.41 | 0.45 | 0.18 | 1.33 | 2.43 | 0.01 | 0.14 | -0.08 | 0.03 | 0.73 | 1.58 | 0.62 | 0.85 |

uses the policy correction $w(a|s)$ as a stationary distribution correction, i.e., $\rho(\pi_w) = \mathbb{E}_{d_D}[w(a|s)r(s,a)]$; **Extraction**, which employs the extracted stationary distribution correction, i.e., $\rho(\pi_w) = \mathbb{E}_{d_D}[w(s)w(a|s)r(s,a)]$, **IHOPE** (Liu et al., 2018), which extracts a stationary distribution from the policy correction while incorporating a discriminator function that adversarially maximizes Bellman-flow constraint violations; and **DualDICE** (Nachum et al., 2019), a representative Q-LP-based OPE algorithm.[2]

The results in Table 1 show that **SemiDICE** consistently underperforms, confirming that the policy correction alone is unsuitable for OPE. In contrast, **Extraction** successfully derives a valid stationary distribution, outperforming the baselines. Compared to the extensively studied Q-LP-based OPE algorithms (e.g., **DualDICE**), our extraction algorithm benefits from in-sample learning, avoiding the risk of evaluating OOD next actions using the function approximator being trained.

---

[2]Notably, our approach based on the policy correction slightly deviates from conventional OPE settings (Nachum et al., 2019; Yang et al., 2020), which typically assume direct access to the policy $\pi(a|s)$. This requires us to extract policy first to apply conventional OPE algorithms.

While **IHOPE** also computes the marginalized correction using the policy correction, its reliance on a min-max optimization framework reduces training stability. In contrast, our algorithm is based on a single convex optimization, ensuring greater stability and superior estimation performance.

### 6.3. Experiments on offline constrained RL

**Setup** Our main offline constrained RL experiment follows the DSRL (Liu et al., 2024) benchmark, comparing algorithm performance across three different environments: Safety-Gymnasium (Marchesini et al., 2021; Ji et al., 2023), Bullet Safety Gym (Gronauer, 2022), and MetaDrive (Li et al., 2022). We compare our **CORSDICE** with: **BC-All**, which imitates the entire dataset, **BC-Safe**, which imitates only safe trajectories, **BCQ-Lag**, a constrained variant of BCQ (Fujimoto et al., 2019) with a PID controller (Stooke et al., 2020), **BEAR-Lag**, a constrained variant of BEAR (Kumar et al., 2019) with a PID controller, and **COptiDICE** (Lee et al., 2021b) (details in Appendix G).

**Results** Table 2 summarizes the results. CORSDICE was the only algorithm to satisfy the cost constraints across all environments, outperforming baselines in 27 out of 38

*Table 3.* Normalized DSRL (Liu et al., 2024) with advanced function approximators and tighter cost limits. The results of baselines with asterisk (*) are adopted from FISOR (Zheng et al., 2024). **Gray**: Unsafe agents, **Bold**: Safe agents whose normalized costs are below 1.0, **Blue**: Safe agents with the highest normalized return.

| TASK | D-BC-ALL | | D-BC-SAFE | | CDT* | | TREBI* | | FISOR* | | D-CORSDICE (OURS) | |
|---|---|---|---|---|---|---|---|---|---|---|---|---|
| | REWARD ↑ | COST ↓ | REWARD ↑ | COST ↓ | REWARD ↑ | COST ↓ | REWARD ↑ | COST ↓ | REWARD ↑ | COST ↓ | REWARD ↑ | COST ↓ |
| CARBUTTON1 | 0.15 | 14.50 | 0.03 | 5.25 | 0.17 | 7.05 | 0.07 | 3.75 | **-0.02** | **0.26** | **-0.02** | **0.90** |
| CARBUTTON2 | 0.11 | 5.32 | -0.02 | 1.12 | 0.23 | 12.87 | **-0.03** | **0.97** | 0.01 | 0.58 | 0.04 | 0.75 |
| CARPUSH1 | 0.21 | 3.66 | 0.15 | 1.34 | 0.27 | 2.12 | 0.26 | 1.03 | **0.28** | **0.28** | 0.24 | 0.50 |
| CARPUSH2 | 0.11 | 2.96 | **0.05** | **0.93** | 0.16 | 4.60 | 0.12 | 2.65 | **0.14** | **0.89** | 0.05 | 0.77 |
| CARGOAL1 | 0.40 | 4.22 | 0.23 | 1.03 | 0.60 | 3.15 | 0.41 | 1.16 | **0.49** | **0.83** | 0.28 | 0.62 |
| CARGOAL2 | 0.34 | 3.67 | 0.15 | 2.35 | 0.45 | 6.05 | 0.13 | 1.16 | 0.06 | 0.33 | **0.11** | **0.59** |
| ANTVEL | 0.98 | 33.12 | 0.68 | 2.16 | **0.98** | **0.91** | 0.31 | 0.00 | 0.89 | 0.00 | 0.91 | 0.58 |
| HALFCHEETAHVEL | 0.93 | 18.73 | **0.73** | **0.25** | **0.97** | **0.55** | 0.87 | 0.23 | 0.89 | 0.00 | 0.87 | 0.02 |
| SWIMMERVEL | 0.45 | 15.08 | **0.45** | **0.82** | 0.67 | 1.47 | 0.42 | 0.31 | -0.04 | 0.00 | 0.12 | 0.84 |
| **SAFETYGYM AVERAGE** | 0.41 | 11.25 | 0.27 | 1.69 | 0.50 | 4.31 | 0.28 | 1.36 | **0.30** | **0.35** | 0.29 | 0.62 |
| ANTRUN | 0.80 | 17.31 | 0.61 | 1.51 | 0.70 | 1.88 | 0.63 | 5.43 | 0.45 | 0.03 | **0.63** | **0.84** |
| BALLRUN | 0.53 | 10.20 | **0.18** | **0.89** | **0.32** | **0.45** | 0.29 | 4.24 | 0.18 | 0.00 | 0.24 | 0.00 |
| CARRUN | 0.90 | 3.37 | **0.86** | **0.44** | 0.99 | 1.10 | 0.97 | 1.01 | **0.73** | **0.14** | **0.93** | **0.57** |
| DRONERUN | 0.60 | 12.08 | 0.48 | 2.75 | **0.58** | **0.30** | 0.59 | 1.41 | 0.30 | 0.55 | 0.55 | 0.32 |
| ANTCIRCLE | 0.55 | 16.89 | 0.41 | 6.04 | 0.48 | 7.44 | 0.37 | 2.50 | 0.20 | 0.00 | **0.34** | **0.23** |
| BALLCIRCLE | 0.73 | 8.76 | **0.13** | **0.28** | 0.68 | 2.10 | 0.63 | 1.89 | 0.34 | 0.00 | **0.40** | **0.26** |
| CARCIRCLE | 0.33 | 10.19 | 0.23 | 1.07 | 0.71 | 2.19 | **0.49** | **0.73** | 0.40 | 0.11 | 0.21 | 0.68 |
| DRONECIRCLE | 0.71 | 9.46 | **0.42** | **0.60** | 0.55 | 1.29 | 0.54 | 2.36 | **0.48** | **0.00** | 0.43 | 0.00 |
| **BULLETGYM AVERAGE** | 0.64 | 11.03 | 0.42 | 1.70 | 0.63 | 2.09 | 0.56 | 2.45 | 0.39 | 0.10 | **0.47** | **0.36** |
| EASYSPARSE | 0.67 | 7.64 | **0.36** | **0.00** | **0.05** | **0.10** | 0.26 | 6.22 | 0.34 | 0.00 | **0.58** | **0.44** |
| EASYMEAN | 0.63 | 7.64 | **0.35** | **0.00** | **0.27** | **0.24** | 0.19 | 4.85 | 0.38 | 0.25 | **0.48** | **0.09** |
| EASYDENSE | 0.54 | 5.84 | **0.33** | **0.00** | 0.43 | 2.31 | 0.26 | 5.81 | 0.36 | 0.25 | **0.59** | **0.31** |
| MEDIUMSPARSE | 0.82 | 5.25 | **0.39** | **0.00** | 0.26 | 2.20 | 0.06 | 1.70 | 0.42 | 0.22 | **0.45** | **0.53** |
| MEDIUMMEAN | 0.84 | 4.63 | **0.53** | **0.01** | 0.28 | 2.13 | 0.20 | 1.90 | 0.39 | 0.08 | 0.45 | 0.53 |
| MEDIUMDENSE | 0.79 | 4.98 | **0.35** | **0.01** | 0.29 | 0.77 | 0.03 | 1.18 | **0.49** | **0.44** | 0.49 | 0.03 |
| HARDSPARSE | 0.49 | 7.04 | **0.36** | **0.00** | 0.17 | 0.47 | 0.00 | 0.82 | 0.30 | 0.01 | 0.25 | 0.18 |
| HARDMEAN | 0.51 | 5.90 | **0.25** | **0.00** | 0.28 | 3.32 | 0.16 | 4.91 | 0.26 | 0.09 | **0.31** | **0.47** |
| HARDDENSE | 0.41 | 4.75 | **0.34** | **0.00** | 0.24 | 1.49 | 0.02 | 1.21 | 0.30 | 0.34 | 0.21 | 0.00 |
| **METADRIVE AVERAGE** | 0.63 | 5.96 | **0.36** | **0.00** | 0.25 | 1.45 | 0.13 | 3.18 | 0.36 | 0.25 | **0.43** | **0.23** |

tasks and achieving the highest average performance. Other baseline methods struggled to balance return maximization and constraint satisfaction, often violating cost constraints or yielding suboptimal returns. This success of CORS-DICE can be attributed to incorporating accurate off-policy cost evaluation into the state-of-the-art semi-gradient DICE methods in unconstrained setting. We performed additional ablation studies on the cost sensitivity in Appendix H.1.

### 6.3.1. WITH ADVANCED FUNCTION APPROXIMATORS

**D-CORSDICE**  Recent research (Chen et al., 2021; Hansen-Estruch et al., 2023; Wang et al., 2022) suggests that using advanced function approximators like transformers (Vaswani, 2017) or diffusion models (Ho et al., 2020; Song et al., 2020) can enhance offline RL performance. Here, we extend CORSDICE by guiding a behavior-cloned diffusion model with our learned policy correction $w(a|s)$, similar to D-DICE (Mao et al., 2024a), and refer to this extension as **D-CORSDICE** (details in Appendix E).

**Setup**  We compare D-CORSDICE with: **D-BC-All**, a behavior-cloning diffusion model, **D-BC-Safe**, diffusion-based BC model trained on safe trajectories, **Constrained Decision Transformer (CDT)** (Liu et al., 2023), **TREBI** (Lin et al., 2023), and **FISOR** (Zheng et al., 2024). We strictly follow the experimental setup of Lin et al. (2023),

who uses a single, tighter cost threshold compared to DSRL benchmark (details in Appendix G).

**Results**  Table 3 summarizes the results. D-CORSDICE and FISOR were the only ones that consistently satisfied cost constraints across all tasks. While both performed comparably in Safety Gymnasium, CORSDICE achieved superior average performance across all environments.

## 7. Conclusion

In this paper, we aimed to extend semi-gradient-based DICE methods, known for their strong performance in offline RL, to the constrained setting. Yet, our findings revealed that semi-gradient DICE algorithms are fundamentally unable to perform policy evaluation, as they produce policy corrections instead of stationary distribution corrections, making their extension to the constrained setting non-trivial. To address this limitation, we proposed a method to recover stationary distribution corrections, and introduced CORSDICE, a novel offline constrained RL algorithm that outperforms existing baselines.

While this work focuses specifically on constrained RL problems, the proposed method can be readily applied to enhance other DICE-based algorithms for problems requiring OPE, such as ROI maximization (Kim et al., 2024b).

# Impact Statement

This paper presents work whose goal is to advance the field of Machine Learning. There are many potential societal consequences of our work, none which we feel must be specifically highlighted here.

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

# A. OptiDICE and COptiDICE

In this section, we provide full derivation of OptiDICE (Lee et al., 2021a) and COptiDICE (Lee et al., 2021b).

## A.1. OptiDICE

We begin with the convex optimization problem (1) OptiDICE solves.

$$\max_{d \geq 0} \mathbb{E}_{(s,a) \sim d}[r(s,a)] - \alpha \sum_{s,a} d_D(s,a) f\left(\frac{d(s,a)}{d_D(s,a)}\right)$$

$$\text{s.t. } (1-\gamma)p_0(s) = \sum_a d(s,a) - \gamma(\mathcal{T}_* d)(s) \ \forall s$$

where $(\mathcal{T}_* d)(s) := \sum_{\bar{s},\bar{a}} T(s \mid \bar{s}, \bar{a}) d(\bar{s}, \bar{a})$. For simplicity of derivation, we reformulate the optimization problem in terms of the stationary distribution correction $w(s,a) = d(s,a)/d_D(s,a)$.

$$\max_{w \geq 0} \mathbb{E}_{(s,a) \sim d_D}[w(s,a)r(s,a)] - \alpha \sum_{s,a} d_D(s,a) f\left(w(s,a)\right)$$

$$\text{s.t. } (1-\gamma)p_0(s) = \sum_a w(s,a)d_D(s,a) - \gamma(\mathcal{T}_* d_w)(s) \ \forall s$$

where $(\mathcal{T}_* d_w)(s) := \sum_{\bar{s},\bar{a}} T(s \mid \bar{s}, \bar{a}) w(\bar{s}, \bar{a}) d_D(\bar{s}, \bar{a})$.

We obtain Lagrangian dual $\max_{w \geq 0} \min_\nu \mathcal{L}(w, \nu)$ of the reformulated problem where $\nu(s)$ is a Lagrangian multiplier for the Bellman flow constraint.

$$\mathcal{L}(w,\nu) := \mathbb{E}_{(s,a) \sim d_D}\left[w(s,a)r(s,a) - \alpha f\left(w(s,a)\right)\right] + \sum_s \nu(s)\left((1-\gamma)p_0(s) + \gamma(\mathcal{T}_*(d_w)(s)) - \sum_a w(s,a)d_D(s,a)\right)$$

$$= \mathbb{E}_{(s,a) \sim d_D}\left[w(s,a)\left(r(s,a) + \gamma \sum_{s'} T(s'|s,a)\nu(s') - \nu(s)\right) - \alpha f\left(w(s,a)\right)\right] + \mathbb{E}_{s_0 \sim p_0}[(1-\gamma)\nu(s_0)]$$

$$= \mathbb{E}_{(s,a) \sim d_D}\left[w(s,a)e_\nu(s,a) - \alpha f\left(w(s,a)\right)\right] + \mathbb{E}_{s_0 \sim p_0}[(1-\gamma)\nu(s_0)] \tag{20}$$

where $\sum_s \nu(s)(\mathcal{T}_* d_w)(s) = \sum_{s,a} w(s,a)d_D(s,a) \sum_{s'} T(s'|s,a)\nu(s')$ and $e_\nu(s,a) = r(s,a) + \gamma \sum_{s'} T(s'|s,a)\nu(s') - \nu(s)$.

Due to the convexity of the problem, strong duality can be established via Slater's condition. We follow the assumption in (Lee et al., 2021a) that all states are reachable within a given MDP. This assumption ensures the strict feasibility of $d(s,a) > 0, \ \forall s,a$, thereby satisfying Slater's condition. The strong duality allows the optimization order to be switched as shown below.

$$\max_{w \geq 0} \min_\nu \mathcal{L}(w,\nu) = \min_\nu \max_{w \geq 0} \mathcal{L}(w,\nu) \tag{21}$$

The reordering enables inner maximization over $w(s,a)$, whose optimal solution satisfies $\frac{\partial \mathcal{L}(w,\nu)}{\partial w(s,a)} = 0 \ \forall s,a$. Optimal $w_\nu^*(s,a)$ can be expressed in a closed form in terms of $\nu$.

$$w_\nu^*(s,a) = \max\left(0, (f')^{-1}\left(\frac{e_\nu(s,a)}{\alpha}\right)\right) \tag{22}$$

When $w_\nu^*(s,a)$ is plugged into the dual function (20), $\nu$ loss of OptiDICE is expressed as,

$$\min_\nu \mathcal{L}(w_\nu^*, \nu) = \mathbb{E}_{s_0 \sim p_0}[(1-\gamma)\nu(s_0)] + \mathbb{E}_{(s,a) \sim d_D}\left[w_\nu^*(s,a)e_\nu(s,a) - \alpha f\left(w_\nu^*(s,a)\right)\right]$$

$$= \mathbb{E}_{s_0 \sim p_0}[(1-\gamma)\nu(s_0)] + \alpha \mathbb{E}_{(s,a) \sim d_D}\left[f_0^*\left(\frac{e_\nu(s,a)}{\alpha}\right)\right] \tag{23}$$

where $f_0^*(y) := \max_{x \geq 0} xy - f(x)$ is a convex conjugate of $f$ in $\mathbb{R}^+$.

**Policy extration** After obtaining the solution $\nu^*$, we need to extract a policy $\pi^*$ that induces the stationary distribution $w_{\nu^*}^*(s,a)d_D(s,a)$. In finite domains, the policy can be computed by $\pi^*(a|s) = \frac{w(s,a)d_D(s,a)}{\sum_a w(s,a)d_D(s,a)}$. In continuous domains, assuming a parameterized policy $\pi_\theta$, we adopt weighted behavior cloning by minimizing:

$$\min_{\pi_\theta} -\mathbb{E}_{(s,a)\sim d_D}[w_{\nu^*}^*(s,a)\log \pi_\theta(a|s)] \tag{24}$$

## A.2. COptiDICE

COptiDICE is a constrained version of OptiDICE, where the following constraint is added to the convex optimization (1):

$$\sum_{s,a} d(s,a)c(s,a) \leq (1-\gamma)C_{\text{lim}} =: \tilde{C}_{\text{lim}}.$$

This results in COptiDICE solving offline constrained RL problem defined as:

$$\max_{w\geq 0} \mathbb{E}_{(s,a)\sim d_D}[w(s,a)r(s,a)] - \alpha\sum_{s,a} d_D(s,a)f(w(s,a))$$

$$\text{s.t. } (1-\gamma)p_0(s) = \sum_a w(s,a)d_D(s,a) - \gamma(\mathcal{T}_* d_w)(s) \ \forall s$$

$$\sum_{s,a} w(s,a)d_D(s,a)c(s,a) \leq \tilde{C}_{\text{lim}}$$

We follow the approach from Appendix A.1 to derive the loss funtions of COptiDICE. We obtain Lagrangian dual $\max_{w\geq 0}\min_{\nu,\lambda\geq 0}\mathcal{L}(w,\nu,\lambda)$ of the reformulated problem where $\lambda$ is additionally introduced as a Lagrangian multiplier for the cost constraint.

$$\mathcal{L}(w,\nu,\lambda) := \mathbb{E}_{(s,a)\sim d_D}[w(s,a)(r(s,a) - \lambda c(s,a) - \alpha f(w(s,a))]$$

$$+ \sum_s \nu(s)\left((1-\gamma)p_0(s) + \gamma(\mathcal{T}_*(d_w)(s)) - \sum_a w(s,a)d_D(s,a)\right) + \lambda\tilde{C}_{\text{lim}} \tag{25}$$

$$= \mathbb{E}_{(s,a)\sim d_D}[w(s,a)e_{\nu,\lambda}(s,a) - \alpha f(w(s,a))] + \mathbb{E}_{s_0\sim p_0}[(1-\gamma)\nu(s_0)] + \lambda\tilde{C}_{\text{lim}} \tag{26}$$

where $e_{\nu,\lambda}(s,a) = r(s,a) - \lambda c(s,a) + \gamma\sum_{s'}T(s'|s,a)\nu(s') - \nu(s)$.

The reordering based on strong duality enables inner maximization over $w(s,a)$, whose optimal solution satisfies $\frac{\partial\mathcal{L}(w,\nu,\lambda)}{\partial w(s,a)} = 0 \ \forall s,a$. Optimal $w_{\nu,\lambda}^*(s,a)$ can be expressed in a closed form in terms of $\nu$ and $\lambda$.

$$w_{\nu,\lambda}^*(s,a) = \max\left(0, (f')^{-1}\left(\frac{e_{\nu,\lambda}(s,a)}{\alpha}\right)\right) \tag{27}$$

When $w_{\nu,\lambda}^*(s,a)$ is plugged into the dual function (26), $\nu$ loss and $\lambda$ loss of COptiDICE can be expressed as,

$$\min_\nu \mathbb{E}_{s_0\sim p_0}[(1-\gamma)\nu(s_0)] + \mathbb{E}_{(s,a)\sim d_D}\left[w_{\nu,\lambda}^*(s,a)e_{\nu,\lambda}(s,a) - \alpha f\left(w_{\nu,\lambda}^*(s,a)\right)\right]$$

$$= \mathbb{E}_{s_0\sim p_0}[(1-\gamma)\nu(s_0)] + \alpha\mathbb{E}_{(s,a)\sim d_D}\left[f_0^*\left(\frac{e_{\nu,\lambda}(s,a)}{\alpha}\right)\right]$$

We derive $\lambda$ loss from (25) to emphasize its role as a Lagrangian multiplier that ensures the satisfaction of the cost constraint:

$$\min_{\lambda\geq 0} \lambda\left(\tilde{C}_{\text{lim}} - \mathbb{E}_{(s,a)\sim d_D}[w_{\nu,\lambda}^*(s,a)c(s,a)]\right)$$

# B. SemiDICE

In this section, we derive the semi-gradient variants of OptiDICE (SemiDICE, f-DVL and ODICE) and clarify the characteristics of their optimal solution. We show that SemiDICE returns a valid policy correction rather than a stationary

distribution correction, while showing f-DVL and ODICE often violates the validity conditions as a policy correction and a stationary distribution correction as depicted in Figure 1. This property makes SemiDICE suitable for CORSDICE framework as it requires a valid policy correction that satisfies $\sum_a w(s,a)\pi_D(a|s) = 1$. While various semi-gradient losses such as dual-V and $f$-DVL were introduced in (Sikchi et al., 2023), we derive SemiDICE due to subtle differences in the loss functions and derivations.

As mentioned in our paper, prior semi-gradient methods have involved three modifications: (1) partially or entirely omitting the gradient from next state $\nu(s')$ in $e_\nu(s,a)$, (2) replacing the initial state distribution, $p_0(s)$, with the dataset distribution, and (3) introducing a hyperparameter $\beta$ to balance loss terms while removing the hyperparameter $\alpha$. We divide the sections based on (1) to separately analyze semi-gradient method that (partially/entirely) omits the gradient from next state $\nu(s')$ in $e_\nu(s,a)$ of $\nu$ loss (23) of OptiDICE.

### B.1. Semi-gradient DICE algorithms that entirely omit the gradient from the next state

We give two semi-gradient DICE algorithms that entirely omit the gradient from the next state $\nu(s')$: SemiDICE and $f$-DVL (Sikchi et al., 2023). They share a common characteristic where the term $r(s,a) + \gamma \sum_{s'} T(s'|s,a)\nu(s')$ within $e_\nu(s,a)$ is separately estimated by an additional function approximator $Q(s,a)$ with $Q$ loss given below:

$$\min_Q \ \mathbb{E}_{(s,a,s')\sim d_D}[(Q(s,a) - (r + \gamma\nu(s')))^2]$$

where the use of $Q(s,a)$ is enabled as the gradient from the next state $\nu(s')$ is completely ignored.

**SemiDICE** We first derive our algorithm, SemiDICE, from $\nu$ loss of OptiDICE (23). We also provide loss function of $f$-DVL for comparison.

$$\mathcal{L}_{\text{OptiDICE}}(\nu) = \sum_s (1-\gamma)p_0(s)\nu(s) + \sum_{s,a} d_D(s,a)\left[\alpha f_0^*\left(\frac{r(s,a) + \gamma\sum_{s'} T(s'|s,a)\nu(s') - \nu(s)}{\alpha}\right)\right] \quad (28)$$

$$\mathcal{L}_{\text{f-DVL}}(\nu) = \sum_{s,a} d_D(s,a)\left[(1-\beta)\nu(s) + \beta f_0^*\left(Q(s,a) - \nu(s)\right)\right] \quad (29)$$

$$\mathcal{L}_{\text{SemiDICE}}(\nu) = \sum_{s,a} d_D(s,a)\left[\nu(s) + \alpha f_0^*\left(\frac{Q(s,a) - \nu(s)}{\alpha}\right)\right] \quad (30)$$

In (Sikchi et al., 2023), $f$-DVL applies the semi-gradient technique to OptiDICE and simply replaces the initial state distribution $p_0(s)$ with the dataset distribution $d_D(s)$. However, we propose an alternative interpretation to demonstrate that the replacement can also be understood as a semi-gradient approach. To establish this, we introduce a minor assumption: the dataset policy $\pi_D(a|s)$ and the policy being optimized $\pi(a|s)$ share the same initial state distribution $p_0(s)$.

Under this assumption, we can replace $(1-\gamma)p_0(s)$ with $-\gamma(\mathcal{T}_* d_D)(s) + \sum_a d_D(s,a)$. This substitution is justified because the stationary distribution of the dataset policy, $d_D(s,a)$, satisfies the Bellman flow constraint as well. Consequently, we extend this relationship to the equality shown below:

$$\sum_s (1-\gamma)p_0(s)\nu(s) = \sum_s \nu(s)\left(-\gamma(\mathcal{T}_* d_D)(s) + \sum_a d_D(s,a)\right) \quad (31)$$

$$= \sum_{s,a} d_D(s,a)\left(-\gamma\sum_{s'} T(s'|s,a)\nu(s') + \nu(s)\right) \quad (32)$$

We apply this relationship to rewrite the Lagrangian dual $\mathcal{L}(w,\nu)$ (20) of OptiDICE.

$$\max_{w\geq 0}\min_\nu \mathcal{L}(w,\nu) = \mathbb{E}_{s\sim p_0}[\nu(s)] + \mathbb{E}_{(s,a)\sim d_D}[w(s,a)e_\nu(s,a) - \alpha f(w(s,a))]$$

$$= \mathbb{E}_{(s,a)\sim d_D}[-\gamma\sum_{s'} T(s'|s,a)\nu(s') + \nu(s)] + \mathbb{E}_{(s,a)\sim d_D}[w(s,a)e_\nu(s,a) - \alpha f(w(s,a))]$$

We follow the same derivation from OptiDICE (Appendix A) to obtain the closed from solution of $w^*(s, a)$ that satisfies $\frac{\partial \mathcal{L}(w, \nu)}{\partial w(s, a)} = 0$. Since the term with initial distribution is independent to the maximization of $w(s, a)$, its closed form solution is equivalent to that of OptiDICE (22).

$$w_\nu^*(s, a) = \max\left(0, (f')^{-1}\left(\frac{r(s, a) + \gamma \sum_{s'} T(s'|s, a)\nu(s') - \nu(s)}{\alpha}\right)\right)$$

We use the closed form solution $w^*(s, a)$ to derive $\nu$ loss without the initial state distribution $p_0(s)$.

$$\min_\nu \ \sum_{s, a} d_D(s, a) \left[-\gamma \sum_{s'} T(s'|s, a)\nu(s') + \nu(s) + \alpha f_0^*\left(\frac{r(s, a) + \gamma \sum_{s'} T(s'|s, a)\nu(s') - \nu(s)}{\alpha}\right)\right] \tag{33}$$

At this point, we apply the semi-gradient technique to neglect the gradients from $\nu(s')$ and approximate $Q(s, a)$ with $r(s, a) + \sum_{s'} T(s'|s, a)\nu(s')$. We note that the gradients $\nu(s')$ were ignored both inside and outside the convex function $f_0^*(x)$.

$$\mathcal{L}_{\text{SemiDICE}}(\nu) = \sum_{s, a} d_D(s, a) \left[\nu(s) + \alpha f_0^*\left(\frac{Q(s, a) - \nu(s)}{\alpha}\right)\right]$$

$$w^*(s, a) = \max\left(0, (f')^{-1}\left(\frac{Q(s, a) - \nu(s)}{\alpha}\right)\right)$$

The semi-gradient technique causes $\nu$ to lose its role as a Lagrangian multiplier that ensures the satisfaction of the Bellman flow constraints of $w^*(s, a)d_D(s, a)$. Proposition 4.1 states that the optimal solution of SemiDICE is a policy correction rather than a valid stationary distribution correction and we provide its proof in the following paragraph.

**Proposition 4.1** The correction $w^*(s, a)$ obtained by the optimal $\nu^* = \arg\min_\nu \mathbb{E}_{d_D}\left[\nu(s) + \alpha f_0^*(\frac{Q(s,a)-\nu(s)}{\alpha})\right]$,

$$w^*(s, a) = \max\left(0, (f')^{-1}\left(\frac{Q(s,a)-\nu^*(s)}{\alpha}\right)\right),$$

violates the Bellman flow constraint (1b) but satisfies the following conditions for $w^*(s, a)$ to act as a policy correction:

$$\sum_a w^*(s, a)\pi_D(a|s) = 1, \ w^*(s, a) \geq 0, \ \forall s, a.$$

*Proof.* The derivative of the $\nu$ loss $\mathcal{L}_{\text{SemiDICE}}(\nu)$ w.r.t. $\nu(s)$ is given as

$$\frac{\partial \mathcal{L}_{\text{SemiDICE}}(\nu)}{\partial \nu(s)} = \sum_a d_D(s, a) \left(1 - (f_0^*)'\left(\frac{Q(s, a) - \nu(s)}{\alpha}\right)\right)$$

$$= \sum_a d_D(s, a) \left(1 - \max\left(0, (f')^{-1}\left(\frac{Q(s, a) - \nu(s)}{\alpha}\right)\right)\right)$$

where $(f_0^*)'(x) = \max(0, (f')^{-1}(x))$.

Due to the convexity of $f_0^*(x)$, optimal $w^*(s, a) = \max\left(0, (f')^{-1}\left(\frac{Q(s,a)-\nu^*(s)}{\alpha}\right)\right)$ obtained from $\nu^*$ that satisfies $\partial \mathcal{L}_{\text{SemiDICE}}(\nu)/\partial \nu = 0$ satisfies the equality below.

$$\sum_a d_D(s, a) \left(1 - w^*(s, a)\right) = 0$$

$$\sum_a w^*(s, a)d_D(s, a) = \sum_a d_D(s, a)$$

$$\sum_a w^*(s, a)\pi_D(a|s) = 1 \ \forall s$$

where state stationary distribution $d_D(s)$ is divided in each sides and $\sum_a \pi_D(a|s) = 1$. This indicates that weighted stationary distribution $w^*(s, a)d_D(s, a)$ is not a valid stationary distribution, while $w^*(s, a)\pi_D(a|s)$ is a policy, as its marginal sum over actions is 1. $\qquad \square$

$f$-**DVL** Based on the derivation of SemiDICE, we identify the optimal solution of $f$-DVL. $\nu$ loss of $f$-DVL $\mathcal{L}_{\text{f-DVL}}$, is simply derived from $\nu$ loss of SemiDICE by reweighting the terms with $\beta$ instead of $\alpha$. $\mathcal{L}_{\text{f-DVL}}$ and its optimal weight $w^*(s, a)$ is given as,

$$\mathcal{L}_{\text{f-DVL}}(\nu) = \sum_{s,a} d_D(s, a) \left[(1 - \beta)\nu(s) + \beta f_0^* \left(Q(s, a) - \nu(s)\right)\right]$$

$$w_{\text{f-DVL}}^*(s, a) = \max \left(0, (f')^{-1} \left(Q(s, a) - \nu(s)\right)\right)$$

We also show the property of the optimal $w_{\text{f-DVL}}^*(s, a)$ from the derivative of $\mathcal{L}_{\text{f-DVL}}$ w.r.t. $\nu(s)$ given as,

$$\frac{\partial \mathcal{L}_{\text{f-DVL}}}{\partial \nu(s)} = \sum_a d_D(s, a) \left(1 - \beta + \beta (f_0^*)' \left(Q(s, a) - \nu(s)\right)\right)$$

$$= \sum_a d_D(s, a) \left(1 - \beta + \beta \max \left(0, (f')^{-1} \left(Q(s, a) - \nu(s)\right)\right)\right)$$

$$= \sum_a d_D(s, a) \left(1 - \beta + \beta w_{\text{f-DVL}}^*(s, a)\right)$$

Optimal correction $w_{\text{f-DVL}}^*$ is obtained from the optimal $\nu^*$ that minimizes $\mathcal{L}_{\text{f-DVL}}$.

$$\beta \sum_a w_{\text{f-DVL}}^*(s, a) d_D(s, a) = (1 - \beta) \sum_a d_D(s, a) \ \forall s$$

$$\sum_a w_{\text{f-DVL}}^*(s, a) \pi_D(a|s) = \frac{1 - \beta}{\beta} \ \forall s$$

This indicates that the weighted stationary distribution $w_{\text{f-DVL}}(s, a) d_D(a, s)$ is not a valid stationary distribution, whereas $w_{\text{f-DVL}}^*(s, a) \pi_D(a|s)$ represents a scaled policy. The weight from $f$-DVL, $w_{\text{f-DVL}}^*(s, a)$, is a valid policy correction only when $\lambda = 0.5$. While the invalidity of $f$-DVL as a policy correction does not impact its offline RL performance, it affects our off-policy evaluation method within Section 5 as it requires a valid policy correction. Therefore, we adopt SemiDICE in our formulation of CORSDICE.

### B.2. Semi-gradient DICE algorithms that partially omits the gradient from the next state

We give a semi-gradient DICE algorithm that partially omits the gradient from the next state $\nu(s')$: ODICE (Mao et al., 2024b). ODICE is an offline RL algorithm built upon $f$-DVL where the gradient from the next state $\nu(s')$ is projected to be orthogonal to the gradient from the current state $\nu(s)$ rather than completely ignoring it.

**ODICE** We demonstrate how ODICE is derived by applying the orthogonal gradient approach to the following loss function similar to $\nu$ loss of $f$-DVL:

$$\min_\nu \sum_{s,a,s'} d_D(s, a, s') \left[(1 - \beta)\nu(s) + \beta f_0^* \left(e_\nu(s, a, s')\right)\right] \tag{34}$$

where $e_\nu(s, a, s') = r(s, a) + \gamma \nu(s') - \nu(s)$. Assuming $\nu$ is parameterized by $\theta$, ODICE addresses the conflict between two gradients within $f_0^* (e_{\nu_\theta}(s, a, s'))$: forward gradient $g_f$ and backward gradient $g_b$.

$$g_f = -(f_0^*)' (e_{\nu_\theta}(s, a, s')) \nabla_\theta \nu_\theta(s)$$

$$g_b = \gamma (f_0^*)' (e_{\nu_\theta}(s, a, s')) \nabla_\theta \nu_\theta(s')$$

The paper claims that the backward gradient may cancel out the effect of the forward gradient, leading to a catastrophic unlearning phenomenon. To address the conflicting gradient issue, the orthogonal gradients are applied to the second term of (34).

$$\nabla_\theta f_0^* (e_{\nu_\theta}(s, a, s')) = g_f + g_b \tag{35}$$

$$\nabla_{\text{ortho}} f_0^* (e_{\nu_\theta}(s, a, s')) = g_f + \eta \left(g_b - \frac{g_b^T g_f}{||g_f||^2} g_f\right) \tag{36}$$

where backward gradient is projected to be orthogonal to the forward gradient, and $\eta$ is a hyperparameter that decides how much the projected gradient is applied. (Mao et al., 2024b) shows that by setting $\eta$ large enough, orthogonal gradient descent can converge to the same point when full gradient is applied, the first term is an expectation over dataset distribution not an expectation over initial state distribution. When (33)

Despite the strong RL performance of ODICE in offline RL, it struggles to converge to a valid stationary distribution correction or policy correction. We identify three key reasons for this issue:

1. Difficulty in choosing an appropriate $\eta$: It is challenging to determine a sufficiently large $\eta$ that ensures the equal convergence point of orthogonal and full gradient descent.

2. Bias in the objective function: ODICE is based on the biased objective (34), where the expectation over the transition probability $T$ appears outside the convex function $f_0^*(x)$. This implies that unless the transition probability is deterministic for all states and actions, the biased objective cannot be directly related to the DICE objectives, where the expectation over $T$ inside $f^*(x)$.

3. Mismatch in the gradients: In (34), the first term replaces the initial state distribution $p_0(s)$ with the dataset distribution $d_D$. This substitution can be interpreted as the application of a semi-gradient method, as demonstrated in the derivation of SemiDICE from (33). Consequently, even if the second term of (34) is updated using its full gradient, neither convergence to policy correction nor satisfaction of the Bellman flow constraints is guaranteed.

Figure 1 provides empirical evidence that ODICE generally fails to converge to either policy correction or stationary distribution correction.

## C. Relationship with Behavior Regularized MDP

In this section, we illustrate the close relationship between SemiDICE, SQL (Xu et al., 2022), and XQL (Garg et al., 2023), all of which address the same Behavior Regularized MDP using different approximation methods. We first present the optimal solution to the Behavior Regularized MDP introduced in (Xu et al., 2022) and elaborate on how practical algorithms are derived from this solution. While SQL and XQL are restricted to specific $f$-divergences (Neyman $\chi^2$-divergence and reverse KL divergence), SemiDICE is not limited to a specific $f$-divergence.

This can be considered an extension of the claim from (Sikchi et al., 2023)—that "XQL is an instance of semi-gradient DICE with initial state distribution replacement, where the $f$-divergence is the reverse KL divergence"—though we extend this in a different manner. Based on Section 4 and Appendix C, we show SemiDICE is an approximation of the behavior-regularized RL.

### C.1. Behavior Regularized MDP

We begin by considering the behavior regularized MDP introduced in (Xu et al., 2022):

$$\max_{\pi} \mathbb{E}\left[\sum_{t=0}^{\infty} \gamma^t \left( r(s_t, a_t) - \alpha f\left(\frac{\pi_D(a_t|s_t)}{\pi(a_t|s_t)}\right)\right)\right] \tag{37}$$

where the reward is penalized with the $f$-divergence between $\pi_D(a|s)$ and $\pi(a|s)$. In prior works, Neyman $\chi^2$-divergence (SQL) and reverse KL divergence (XQL) between $\pi_D(a|s)$ and $\pi(a|s)$ were employed. We obtain an equivalent MDP by replacing $f(x)$ with $g(x) = xf(1/x)$, as follows:

$$\max_{\pi} \mathbb{E}\left[\sum_{t=0}^{\infty} \gamma^t \left( r(s_t, a_t) - \alpha \frac{\pi_D(a_t|s_t)}{\pi(a_t|s_t)} f\left(\frac{\pi(a_t|s_t)}{\pi_D(a_t|s_t)}\right)\right)\right] \tag{38}$$

This results in reversing the order of the $f$-divergence, where the reward is penalized with the corresponding $f$-divergence between $\pi(a|s)$ and $\pi_D(a|s)$.

**Reverse relationship between $f$-divergences** If $D_f(\pi||\pi_D)$ is the $f$-divergence between $\pi$ and $\pi_D$, then $D_g(\pi||\pi_D)$ characterized by $g(x) = xf(1/x)$ is also an $f$-divergence and $D_f(\pi||\pi_D) = D_g(\pi_D||\pi)$. We first show $g(x)$ is a valid function for $f$-divergence if $f(x)$ is valid. We list three properties that $f(x)$ satisfies as a valid function for $f$-divergence.

1. Convexity of $f(x)$ in its domain:

    • $f(\theta x + (1-\theta)y) \leq \theta f(x) + (1-\theta)f(y)$, with $0 \leq \theta \leq 1$, $\forall x, y \in$ dom $f$

2. $f(1) = 0$ and strict convexity of $f(x)$ at 1

    • If $\theta x + (1-\theta)y = 1$, $f(1) = f(\theta x + (1-\theta)y) < \theta f(x) + (1-\theta)f(y)$, with $0 \leq \theta \leq 1$, $\forall x, y \in$ dom $f$

We show that $g(x) = xf(1/x)$ also satisfies these properties by using the properties of $f$. We define $c = \theta x + (1-\theta)y$.

1. Convexity of $g(x)$ in its domain:

    • $g(\theta x + (1-\theta)y) = cf(1/c) = cf(\frac{\theta x}{c}\frac{1}{x} + \frac{(1-\theta)y}{c}\frac{1}{y}) \leq \theta g(x) + (1-\theta)g(y)$, with $0 \leq \theta \leq 1$, $\forall x, y \in$ dom $g$

2. $g(1) = 0$ and strict convexity of $g(x)$ at 1

    • $g(1) = f(1) = 0$
    • If $c = 1$, $g(1) = f(1) = f(\frac{\theta x}{1}\frac{1}{x} + \frac{(1-\theta)y}{1}\frac{1}{y}) < \theta g(x) + (1-\theta)g(y)$, with $0 \leq \theta \leq 1$, $\forall x, y \in$ dom $g$

As $f$-divergence characterized by $g(x)$ is a valid $f$-divergence, we now show their reverse relationship.

$$D_f(\pi||\pi_D) := \sum_{s,a} \pi_D(a|s)f\left(\frac{\pi(a|s)}{\pi_D(a|s)}\right) = \sum_{s,a} \pi_D(a|s)\frac{\pi(a|s)}{\pi_D(a|s)}f\left(\frac{\pi_D(a|s)}{\pi(a|s)}\right) = D_g(\pi_D||\pi)$$

## C.2. Optimal solution of Behavior Regularized MDP

We provide a proof on Proposition 4.2 by deriving the optimal policy $\pi^*(a|s)$, and its corresponding value functions for our behavior regularized MDP (38). Following the derivations of (Xu et al., 2022), the policy evaluation operator $\mathcal{T}_f^\pi$ of the behavior regularized MDP is given by,

$$(\mathcal{T}_f^\pi Q)(s,a) := r(s,a) + \gamma\mathbb{E}_{s'\sim T(\cdot|s,a)}[V(s')]$$

$$V(s) = \mathbb{E}_{a\sim\pi(\cdot|s)}\left[Q(s,a) - \alpha\frac{\pi_D(a|s)}{\pi(a|s)}f\left(\frac{\pi(a|s)}{\pi_D(a|s)}\right)\right]$$

The policy learning objective can be expressed as $\max_\pi \mathbb{E}_{s\sim d_D}[V(s)]$, where $D$ denotes the dataset distribution. Accordingly, we formulate a convex optimization problem that optimizes policy $\pi$ by solving $\max_\pi \mathbb{E}_{s\sim D}[V(s)]$, given $Q(s,a)$ within the dataset $d_D$. We add the constraints to ensure the optimized policy $\pi(a|s)$ is a valid policy.

$$\max_\pi \sum_{s,a} d_D(s)\pi(a|s)Q(s,a) - \alpha\sum_{s,a} d_D(s)\pi_D(a|s)f\left(\frac{\pi(a|s)}{\pi_D(a|s)}\right)$$

$$\text{s.t. } \sum_a \pi(a|s) = 1 \ \forall s, a$$

$$\pi(a|s) \geq 0 \ \forall s, a$$

To solve the convex optimization problem, we derive the Lagrangian dual with Lagrangian multiplier $U(s)$ and $\beta(s,a)$ for each policy constraints.

$$\max_\pi \min_{U,\beta\geq 0} L(\pi, U, \beta) = \sum_{s,a} d_D(s)\pi(a|s)Q(s,a) - \alpha\sum_s d_D(s)\pi_D(a|s)f\left(\frac{\pi(a|s)}{\pi_D(a|s)}\right)$$

$$- \sum_s d_D(s)\sum_a U(s)\left(\pi(a|s) - 1\right) - \beta(s,a)\pi(a|s)$$

The KKT conditions of the problem are as follows,

$$\pi^*(a|s) \geq 0, \ \forall s, a \text{ and } \sum_a \pi^*(a|s) = 1, \ \forall s$$

$$\beta^*(s,a) \geq 0, \ \forall s, a$$

$$\beta^*(s,a)\pi^*(a|s) = 0, \ \forall s, a$$

$$Q(s,a) - \alpha f'\left(\frac{\pi^*(a|s)}{\pi_D(a|s)}\right) - U^*(s) - \beta^*(s,a) = 0, \ \forall s, a \tag{39}$$

Due to the sationarity condition (39), the optimal policy correction $w^*(a|s)$ given $U^*(s)$ is

$$\pi^*(a|s) = \max\left(0, (f')^{-1}\left(\frac{Q(s,a) - U^*(s)}{\alpha}\right)\right)\pi_D(a|s), \ \forall s, a \tag{40}$$

We emphasize that the optimality of $U^*(s)$ is independent to the optimality of $Q(s,a)$. Even if $Q(s,a)$ is not equivalent to the optimal $Q^*(s,a)$ of (38), $\pi^*(a|s)$ derived from $U^*(s)$ is still a valid policy correction, due to the role of $U(s)$ as Lagrangian multiplier.

To formulate a loss function solely on $U(s)$, we switch the order of optimization based on strong duality ($\max_\pi \min_{U,\beta \geq 0} L(\pi, U, \beta) = \min_{U,\beta \geq 0} \max_\pi L(\pi, U, \beta)$). Slater's condition for the strong duality is easily satisfied as there exists a policy that satisfies $\pi(a|s) > 0, \forall s, a$. We then insert the optimal solutions $\pi^*(a|s)$ and $\beta^*(s,a)$ into $L(\pi, U, \beta)$ which results in:

$$\min_U L(\pi^*, U, \beta^*) = \sum_s d_D(s)\left[U(s) + \sum_a \pi^*(a|s)(Q(s,a) - U(s)) - \alpha \sum_a \pi_D(a|s)f\left(\frac{\pi^*(a|s)}{\pi_D(a|s)}\right)\right]$$

$$= \mathbb{E}_{(s,a)\sim d_D}\left[U(s) + \alpha f_0^*\left(\frac{Q(s,a) - U(s)}{\alpha}\right)\right] \tag{41}$$

The optimal Lagrangian dual $L(\pi^*, U^*, \beta^*)$ is equivalent to the optimal solution $V^*(s) = \max_\pi \mathbb{E}_{s \sim D}[V(s)]$ given $Q(s,a)$.

$$V^*(s) = U^*(s) + \mathbb{E}_{a \sim \pi_D(\cdot|s)}\left[\alpha f_0^*\left(\frac{Q(s,a) - U^*(s)}{\alpha}\right)\right], \ \forall s$$

**Proposition 4.2** Therefore, in the behavior regularized MDP (38), the optimal value functions $Q^*(s,a)$ and $V^*(s)$ and its corresponding optimal policy $\pi^*$ satisfy the following optimality conditions for all states and actions.

$$U^*(s) = \arg\min_{U(s)} U(s) + \mathbb{E}_{a \sim \pi_D(\cdot|s)}\left[\alpha f_0^*(\frac{Q^*(s,a) - U(s)}{\alpha})\right] \tag{42a}$$

$$V^*(s) = U^*(s) + \mathbb{E}_{a \sim \pi_D(\cdot|s)}\left[\alpha f_0^*(\frac{Q^*(s,a) - U^*(s)}{\alpha})\right] \tag{42b}$$

$$Q^*(s,a) = r(s,a) + \gamma \mathbb{E}_{s' \sim T(\cdot|s,a)}[V^*(s')] \tag{42c}$$

$$\pi^*(a|s) = \max\left(0, (f')^{-1}\left(\frac{Q^*(s,a) - U^*(s)}{\alpha}\right)\right)\pi_D(a|s) \tag{42d}$$

We now demonstrate how the optimal solution of the behavior regularized MDP (42) is approximated by SemiDICE and SQL. We also demonstrate a special case, XQL, that does not require any approximation.

**Approximation in SemiDICE**  We show that SemiDICE approximates the optimal solution of the behavior-regularized MDP (42) by eliminating $V$ and approximating $V^*$ with $U^*$, i.e., $V^*(s) \approx U^*(s)$. To elaborate, we give the loss functions

and the optimal policy of SemiDICE.

$$\min_{\nu} \mathbb{E}_{(s,a)\sim d_D}\left[\nu(s) + \alpha f_0^*\left(\frac{Q^*(s,a) - \nu(s)}{\alpha}\right)\right] \tag{43a}$$

$$\min_{Q} \mathbb{E}_{(s,a)\sim d_D}\left[(r(s,a) + \gamma\nu(s') - Q(s,a))^2\right] \tag{43b}$$

$$\pi^*(a|s) = \max\left(0, (f')^{-1}\left(\frac{Q^*(s,a) - \nu^*(s)}{\alpha}\right)\right)\pi_D(a|s) \tag{43c}$$

where $U^*$ of the behavior regularized MDP (42a) and $\nu^*(s)$ of (43a) are equivalent as they converge to same value given $Q(s,a)$. SemiDICE omits the computation of $V$ (42b) and uses only $\nu$ to update $Q$, which is equivalent to approximating $\mathbb{E}_{a\sim\pi_D(\cdot|s)}\left[\alpha f_0^*(\frac{Q^*(s,a)-U^*(s)}{\alpha})\right]$ with 0. This indicates that optimal $Q^*(s,a)$ of SemiDICE is an approximation of optimal $Q^*(s,a)$ of behavior regularized MDP. However, we emphasize that (43c) is still a valid policy correction as optimization on $\nu$ acted equivalently to the Lagrangian multiplier $U$ that ensures the satisfaction of policy constraints.

**Approximation in SQL**  We show that SQL approximates the optimal solution of the behavior-regularized MDP (42) by eliminating $U$ and approximating $U^*$ with $V^*$, i.e., $U^*(s) \approx V^*(s) - \alpha$. Before describing the approximation, we first apply the $f$-divergence used in SQL to the behavior regularized MDP (42): Neyman $\chi^2$-divergence between $\pi_D(a|s)$ and $\pi(a|s)$ ($g(x) = 1/x + 1$), which is equivalent to $\chi^2$-divergence between $\pi(a|s)$ and $\pi_D(a|s)$ ($f(x) = x^2 - x$).

$$\pi^*(a|s) = \max\left(0, \frac{1}{2} + \frac{Q^*(s,a) - U^*(s)}{2\alpha}\right)\pi_D(a|s) \tag{44a}$$

$$U^*(s) = \arg\min_{U(s)} U(s) + \mathbb{E}_{a\sim\pi_D(\cdot|s)}\left[\alpha\max\left(0, \frac{1}{2} + \frac{Q^*(s,a) - U(s)}{2\alpha}\right)\left(\frac{1}{2} + \frac{Q^*(s,a) - U(s)}{2\alpha}\right)\right] \tag{44b}$$

$$V^*(s) = U^*(s) + \mathbb{E}_{a\sim\pi_D(\cdot|s)}\left[\alpha\max\left(0, \frac{1}{2} + \frac{Q^*(s,a) - U^*(s)}{2\alpha}\right)\left(\frac{1}{2} + \frac{Q^*(s,a) - U^*(s)}{2\alpha}\right)\right] \tag{44c}$$

$$Q^*(s,a) = r(s,a) + \gamma\mathbb{E}_{s'\sim T(\cdot|s,a)}[V^*(s')] \tag{44d}$$

where $f_0^*(y) = \max\left(0, \frac{1+y}{2}\right)\left(\frac{1+y}{2}\right)$. By applying (44a) to (44c), the following equality is satisfied:

$$V^*(s) = U^*(s) + \alpha\mathbb{E}_{a\sim\pi_D(\cdot|s)}\left[\left(\frac{\pi^*(a|s)}{\pi_D(a|s)}\right)^2\right] \approx U^*(s) + \alpha$$

where the second term is approximated to $\alpha$ in SQL. The approximation leads to the replacement of $U(s)$ within (44b) with $V(s) - \alpha$, which leads to the loss functions and the optimal policy of SQL given by:

$$\min_{V} \mathbb{E}_{(s,a)\sim d_D}\left[V(s) + \alpha\max\left(0, 1 + \frac{Q^*(s,a) - V(s)}{2\alpha}\right)\left(1 + \frac{Q^*(s,a) - V(s)}{2\alpha}\right)\right]$$

$$\min_{Q} \mathbb{E}_{(s,a)\sim d_D}\left[(r(s,a) + \gamma\nu(s') - Q(s,a))^2\right]$$

$$\pi^*(a|s) = \max\left(0, 1 + \frac{Q^*(s,a) - V^*(s)}{2\alpha}\right)\pi_D(a|s)$$

However, the approximation causes $V(s)$ of SQL to converge to $U^*(s) + \alpha$ from $U^*(s)$ (44b), rather than $V^*(s)$ (44c). We introduce two special cases where the equality between $U^*(s) + \alpha = V^*(s)$ is satisfied.

**Special case in XQL**  We show that XQL converges to the optimal solution of the behavior-regularized MDP with reverse KL divergence betweeen $\pi_D$ and $\pi$ ($g(x) = -\log x$), which is equivalent to KL-divergence between $\pi$ and $\pi_D$

$(f(x) = x \log x)$. We apply the $f$-divergence to the behavior regularized MDP (42):

$$\pi^*(a|s) = \exp\left(\frac{Q^*(s,a) - U^*(s)}{\alpha} - 1\right)\pi_D(a|s) \tag{45a}$$

$$U^*(s) = \arg\min_{U(s)} U(s) + \mathbb{E}_{a \sim \pi_D(\cdot|s)}\left[\alpha \exp\left(\frac{Q^*(s,a) - U(s)}{\alpha} - 1\right)\right] \tag{45b}$$

$$V^*(s) = U^*(s) + \mathbb{E}_{a \sim \pi_D(\cdot|s)}\left[\alpha \exp\left(\frac{Q^*(s,a) - U^*(s)}{\alpha} - 1\right)\right] \tag{45c}$$

$$Q^*(s,a) = r(s,a) + \gamma \mathbb{E}_{s' \sim T(\cdot|s,a)}[V^*(s')] \tag{45d}$$

where $f_0^*(y) = \exp(y - 1)$. By applying (45a) to (45c), the following equality is satisfied:

$$V^*(s) = U^*(s) + \alpha \mathbb{E}_{a \sim \pi_D(\cdot|s)}\left[\frac{\pi^*(a|s)}{\pi_D(a|s)}\right] = U^*(s) + \alpha$$

where the approximations of the previous algorithms are not required. The loss functions and the optimal policy of XQL are obtained by substituting $U(s)$ with $V(s) - \alpha$:

$$\min_V \mathbb{E}_{(s,a) \sim d_D}\left[V(s) + \alpha \exp\left(\frac{Q(s,a) - V(s)}{\alpha}\right)\right]$$

$$\min_Q \mathbb{E}_{(s,a) \sim d_D}\left[(r(s,a) + \gamma \nu(s') - Q(s,a))^2\right]$$

$$\pi^*(a|s) = \exp\left(\frac{Q^*(s,a) - V^*(s)}{\alpha}\right)\pi_D(a|s)$$

While XQL solves the behvaior regularized MDP with no approximation, the exponential term within $V$ loss makes the algorithm prone to divergence. While the instability can be avoided by adopting high $\alpha$, the regularization becomes to strong and causes its performance of $\pi$ to be bound to $\pi_D$.

### C.3. Proof on SemiDICE avoiding the sparsity problem

We show SemiDICE and other behavior-regularized does not suffer from the sparsity problem OptiDICE suffers by providing the proof below:

**Corollary 4.3** Let $w^*$ be the correction optimized by running SemiDICE. There is no state $s$ where $w^*(s,a) = 0\ \forall a$.

*Proof.* Assume there exists a state $s$ whose $w^*(s,a) = 0\ \forall a$. The assumption contradicts $\sum_a w^*(s,a)\pi_D(a|s) = 1\ \forall a$ as $\sum_a w^*(s,a)\pi_D(a|s) = 0$ in the state $s$, therefore SemiDICE does not suffer from the sparsity problem. $\qquad\square$

## D. State stationary distribution extraction

In this section, we provide a detailed derivation on state stationary extraction (**Extraction**), where we obtain state stationary distribution correction $w(s)$ induced by policy correction $w(a|s)$. After obtaining $w(s)$, stationary distribution $d(s,a) = w(s)w(a|s)d_D(s,a)$ can be utilized for off-policy cost evaluation in offline constrained RL.

$$\mathbb{E}_{(s,a) \sim d}[c(s,a)] = \mathbb{E}_{(s,a) \sim d_D}[w(s)w(a|s)c(s,a)]$$

We formulate a novel convex optimization problem whose optimal solution corresponds to the state stationary distribution ratio, $w(s)$. We assume that the policy correction, $w(a|s)$, is given:

$$\max_{w(s) \geq 0} -\sum_s d_D(s)f(w(s)) \tag{46a}$$

$$\text{s.t. } w(s)d_D(s) = (1-\gamma)p_0(s) + \gamma(\mathcal{T}_* d_w)(s)\ \forall s \tag{46b}$$

where $(\mathcal{T}_* d_w)(s) := \sum_{\bar{s},\bar{a}} T(s \mid \bar{s}, \bar{a})w(\bar{s})w(\bar{a}|\bar{s})d_D(\bar{s}, \bar{a})$.

Regardless of the objective, the $|S|$ Bellman flow constraints in the problem is sufficient to uniquely determine $w(s)$. However, the $f$-divergence between $w(s)d_D(s)$ and $d_D(s)$ introduces convexity into optimization, enabling the application of convex optimization and efficient sample-based optimization.

The Lagrangian dual of the problem, with Lagrange multipliers $\mu(s)$ for the constraint (46b), is given as:

$$\max_{w(s) \geq 0} \min_{\mu} \mathcal{L}(w, \mu) := -\sum_s d_D(s) f(w(s)) + \sum_s \mu(s) \left((1 - \gamma)p_0(s) + \gamma(\mathcal{T}_* d_w)(s) - w(s)d_D(s)\right)$$

$$= \sum_s (1 - \gamma)p_0(s)\mu(s) + \sum_{s,a} d_D(s, a) \left( w(s)w(a|s) \left( \gamma \sum_{s'} T(s'|s, a)\mu(s') - \mu(s) \right) - f(w(s)) \right) \quad (47)$$

$$= (1 - \gamma)\mathbb{E}_{s_0 \sim p_0}[\mu(s_0)] + \mathbb{E}_{(s,a) \sim d_D}[w(s)w(a|s)e_\mu(s, a) - f(w(s))] \quad (48)$$

where $e_\mu(s, a) = \gamma \sum_{s'} T(s'|s, a)\mu(s') - \mu(s)$, and (47) is derived by using the following equality:

$$\sum_s \mu(s)(\mathcal{T}_* d_w)(s)) = \sum_s \mu(s) \sum_{\bar{s}, \bar{a}} T(s|\bar{s}, \bar{a})w(\bar{s})w(\bar{a}|\bar{s})d_D(\bar{s}, \bar{a})$$

$$= \sum_{s'} \mu(s') \sum_{s,a} T(s'|s, a)w(s)w(a|s)d_D(s, a)$$

$$= \sum_{s,a} w(s)w(a|s)d_D(s, a) \sum_{s'} T(s'|s, a)\mu(s')$$

Following the assumption of OptiDICE, the strong duality holds by satisfying Slater's condition, which enables the optimization order to be switched as shown below:

$$\max_{w(s) \geq 0} \min_{\mu} \mathcal{L}(w, \mu) = \min_{\mu} \max_{w \geq 0} \mathcal{L}(w, \mu)$$

The reordering enables inner maximization over $w(s)$, whose optimal solution satisfies $\frac{\partial \mathcal{L}(w, \mu)}{\partial w(s)} = 0 \ \forall s$. Optimal $w_\mu^*(s)$ can be expressed in a closed form in terms of $\mu$.

$$w_\mu^*(s) = \max(0, (f')^{-1}(\mathbb{E}_{a \sim \pi_D}[w(a|s)e_\mu(s, a)])) \quad (49)$$

When $w_\mu^*(s)$ is plugged into the dual function (48), $\mu$ loss of **Extraction** is expressed as,

$$\min_{\mu} \mathcal{L}_{\text{ext}}(\mu) := (1 - \gamma)\mathbb{E}_{s_0 \sim p_0}[\mu(s_0)] + \mathbb{E}_{s \sim d_D}[f_0^*(\mathbb{E}_{a \sim \pi_D}[w(a|s)e_\mu(s, a)])] \quad (50)$$

However, sample-based optimization on $\min_{\mu} \mathcal{L}_{\text{ext}}(\mu)$ is challenging due to the existence of expectations over the transition probability $T$ within $e_\mu(s, a)$ and the dataset policy $\pi_D$ inside the convex function $f_0^*(x)$. Using a naive single-sample estimate such as $\mathbb{E}_{(s,a,s') \sim d_D}[f_0^*(w(a|s)(\gamma\mu(s') - \mu(s)))]$ results in significant bias.

To circumvent this bias issue, we propose a simple bias reduction technique by incorporating an additional function approximator, $A(s)$, to estimate the expectation inside $f_0^*(\cdot)$. We then decompose the $\mu$ optimization of (50) into the following optimizations on $A$ and $\mu$, which share the same optimal solution of $\mu$:

$$\min_A \mathbb{E}_{(s,a,s') \sim d_D}\left[\left(A(s) - w(a|s)\hat{e}_\mu(s, s')\right)^2\right] \quad (51a)$$

$$\min_{\mu} \tilde{\mathcal{L}}_{\text{ext}}(\mu) := (1 - \gamma)\mathbb{E}_{s_0 \sim p_0}[\mu(s_0)] + \mathbb{E}_{(s,a,s') \sim d_D}[(f_0^*)'(A(s))w(a|s)\hat{e}_\mu(s, s')], \quad (51b)$$

where $\hat{e}_\mu(s, s') = \gamma\mu(s') - \mu(s)$.

**Proposition 5.1** Minimization of the objectives in (50) results in the same optimal $\mu^*$ as in (51).

*Proof.* We show optimal $A^*(s)$ of (51a) given $\mu$.

$$A^*(s) = \mathbb{E}_{(a,s')\sim d_D}[w(a|s)\,(\hat{e}_\mu(s,s'))]$$
$$= \mathbb{E}_{a\sim\pi_D}[w(a|s)\mathbb{E}_{s'\sim T}\,[\hat{e}_\mu(s,s')]]$$
$$= \mathbb{E}_{a\sim\pi_D}[w(a|s)e_\mu(s,a)],\ \forall s$$

For simplicity in expression, we assume $\mu(s)$ is parameterized by $\theta$. We show the gradient of $\mathcal{L}_{\text{ext}}(\mu_\theta)$ and $\tilde{\mathcal{L}}_{\text{ext}}(\mu_\theta)$ are the same given $A^*(s) = \mathbb{E}_{a\sim\pi_D}[w(a|s)e_{\mu_\theta}(s,a)]$.

$$\frac{\partial}{\partial\theta}\mathcal{L}_{\text{ext}}(\mu_\theta) = \frac{\partial}{\partial\theta}\tilde{\mathcal{L}}_{\text{ext}}(\mu_\theta)$$

We compare the gradients of the second term of $\mathcal{L}_{\text{ext}}(\mu_\theta)$ and $\tilde{\mathcal{L}}_{\text{ext}}(\mu_\theta)$

$$\frac{\partial}{\partial\theta}\left(\mathbb{E}_{s\sim d_D}\left[f_0^*(\mathbb{E}_{a\sim\pi_D}[w(a|s)e_{\mu_\theta}(s,a)])\right]\right) = \mathbb{E}_{s\sim d_D}\left[(f_0^*)'(A^*(s))\frac{\partial K_{\mu_\theta}(s)}{\partial\theta}\right]$$
$$= \mathbb{E}_{s\sim d_D}\left[(f_0^*)'(A^*(s))\mathbb{E}_{a\sim\pi_D}\left[w(a|s)\frac{\partial(\gamma\mu_\theta(s')-\mu_\theta(s))}{\partial\theta}\right]\right]$$
$$= \mathbb{E}_{(s,a)\sim d_D}\left[(f_0^*)'(A^*(s))w(a|s)\frac{\partial(\gamma\mu_\theta(s')-\mu_\theta(s))}{\partial\theta}\right]$$
$$= \frac{\partial}{\partial\theta}\left(\mathbb{E}_{(s,a)\sim d_D}[(f_0^*)'(A^*(s))w(a|s)\hat{e}_{\mu_\theta}(s,s')]\right)$$

where $K_{\mu_\theta}(s) = \mathbb{E}_{(a,s')\sim d_D}[w(a|s)\,(\gamma\mu_\theta(s')-\mu_\theta(s))]$, and $\frac{\partial K_{\mu_\theta}(s)}{\partial\theta} = \mathbb{E}_{(a,s')\sim d_D}\left[w(a|s)\frac{\partial(\gamma\mu_\theta(s')-\mu_\theta(s))}{\partial\theta}\right]$. The equivalence leads to $\frac{\partial}{\partial\theta}\tilde{\mathcal{L}}_{\text{ext}}(\mu_\theta) = \frac{\partial}{\partial\theta}\mathcal{L}_{\text{ext}}(\mu_\theta)$. Therefore, minimization of (50) results in the same optimal $\mu^*$ as in (51). $\qquad\square$

After the optimization on $A$ and $\mu$, the state stationary distribution correction $w(s)$, corresponding to policy correction $w(a|s)$, is obtained by substituting $A^*(s)$ into (49),

$$w(s) = \max\left(0, (f')^{-1}(A^*(s))\right),\ \forall s$$

# E. D-CORSDICE

In Section 6.3.1, we introduced an extended verion of CORSDICE, which utilizes diffusion (Ho et al., 2020; Yang et al., 2020)-based policy to compare against other baselines that adopts advanced function approximators for actor. While the most of implementation follows that of D-DICE (Mao et al., 2024a), we re-state the objective function and architectural choices for the completeness.

The main contribution of D-DICE was to introduce two way of utilizing a diffusion model in DICE framework, namely *guide* and *select*. Given the pre-trained behavior cloning diffusion model, *guide* method, as the name suggests, guides the denoising process of stochastic differential equation (SDE)-based diffusion models with the learned correction $w(a_0 \mid s)$:

$$\nabla_{a_t}\log\pi_t(a_t \mid s) = \nabla_{a_t}\log\pi_t^D(a_t \mid s) + \tau\cdot\nabla_{a_t}\log\mathbb{E}_{a_0\sim\pi^D(a_0|a_t,s)}[w(a_0 \mid s)]$$

where $\tau$ is a hyperparameter for scaling the guidance score, $\pi^D$ is a behavior cloned model, and $t$ is a timestep of denoising process, not MDP. Note that the compared to Eq. (6) in (Mao et al., 2024a), the stationary distribution correction is replaced with the policy correction. This is because, while D-DICE stated using stationary distribution correction to guide the diffusion model, what actually used was the policy correction, as they are using semi-gradient DICE methods and semi-gradient DICE methods extract policy correction 4.1. However, this error does not invalidate the original D-DICE, as the relationship $\pi^*(a \mid s) = w(a \mid s)\pi^D(a \mid s)$, still holds.

In *select* period, with the guided sampling, we sample multiple actions for a given state. Then, utilizing a learned $Q$ function from semi-gradient DICE method, we can choose a greedy action that maximizes $Q$-value. Two methods combined, they are called *guide-then-select* method, and we adopted this method accordingly.

### E.1. Choice of $f$-divergence

As stated in D-DICE (Mao et al., 2024a), the choice of $f$-divergence affects the stability of training diffusion model. We used Soft-$\chi^2$ divergence, introduced in OptiDICE (Lee et al., 2021a) and defined as:

$$f_{\text{soft-}\chi^2}(x) := \begin{cases} \frac{1}{2}(x-1)^2 & x \geq 1 \\ x \log x - x + 1 & 0 \leq x < 1 \end{cases}$$

D-DICE used slightly different version of $f$, where $x \geq 1$ part is replaced with $(x-1)^2$. Since the difference was minor, we used the Soft-$\chi^2$ as we did in the non-diffusion CORSDICE experiment.

### E.2. Choice of Diffusion Model

We used SDE-based diffusion model (Yang et al., 2020), where the forward process is defined as:

$$d\mathbf{x} = f(\mathbf{x}, t)dt + g(t)d\mathbf{w}$$

where $\mathbf{x}_0 \sim p_0$ and $\mathbf{w}$ is a Brownian motion. (Yang et al., 2020) demonstrated that given an arbitrary drift coefficient $f(\cdot, t) : \mathbb{R}^d \to \mathbb{R}$ and a diffusion coefficient $g(\cdot) : \mathbf{R} \to \mathbf{R}$, there exists an corresponding *reverse* process of generating samples from the noisy data.

While the choice of drift and diffusion coefficients can be arbitrary, we adopt the Variance Preserving (VP) SDE proposed in (Yang et al., 2020) with the linear noise scheduling, given by:

$$d\mathbf{x} = -\frac{1}{2}\beta(t)\mathbf{x}dt + \sqrt{\beta(t)}d\mathbf{w}$$
$$\beta(t) = \beta_{\min} + t(\beta_{\max} - \beta_{\min})$$

For the choice of hyperparameters and network architecutres, please refer to Appendix G.

## F. Details of Section 6.1

### F.1. Finite MDP experiment

We validate our algorithm **SemiDICE** and **Extraction** by following the experimental protocol of (Laroche et al., 2019; Lee et al., 2020). We repeat the experiment for 300 times and average the reults. In each run, an MDP with $|S| = 30, |A| = 4, \gamma = 0.95$ is randomly generated. Initial probability $p_0(s)$ is set to be deterministic for a fixed state. We set there are four possible next states for each state-action pairs, and generate transition probability $T(s'|s, a)$ from Dirichlet distribution $[p(s_1|s, a), p(s_2|s, a), p(s_3|s, a), p(s_4|s, a)] \sim \text{Dir}(1, 1, 1, 1)$ for every state-action pairs $(s, a)$. The reward of 1 is given to a state a single goal state that minimizes the optimal state value at the initial state; other states have zero rewards.

Assuming an offline setting, the dataset policy $\pi_D(a|s)$ is obtained by the mixture of optimal policy $\pi^*$ of the generated MDP and uniformly random policy $\pi_{\text{unif}}$, where $\pi_D(a|s) = 0.5\pi^*(a|s) + 0.5\pi_{\text{unif}}(a|s) \; \forall s, a$. Then 30 trajectories are collected using the generated MDP and the dataset policy $\pi_D(a|s)$. Finally, we construct MLE MDP $\hat{\mathcal{M}} = \langle S, A, T_{\text{mle}}, r, p_0, \gamma \rangle$ using the offline dataset, then test the following algorithms: four DICE-based RL algorithms (**OptiDICE**, **SemiDICE**, **f-DVL**, **ODICE**), two behvaior-regularized RL algorithms (**SQL**, **XQL**), and **Extraction**, which applies the state stationary distribution extraction method to **SemiDICE**. We test the four algorithms (**OptiDICE**, **SemiDICE,SQL, XQL**) over 6 different $\alpha \in \{0.0001, 0.001, 0.01, 0.1, 1.0, 10.0\}$ and two algorithms (**f-DVL**, **ODICE**) over 6 different $\beta \in \{0.1, 0.3, 0.5, 0.7, 0.9, 0.99\}$. Additional hyperparameter for **ODICE** is set to $\eta = 1.0$. We note **Extraction** does not require any hyperparameters. For the choice of $f$-divergence, we adopt $\chi^2$-divergence ($f(x) = \frac{1}{2}(x-1)^2$) for SemiDICE and KL-divergence($f(x) = x \log x$) for state stationary distribution extraction. We note that state stationary distribution extraction returns the same $w(s)$ regardless of the choice of the $f$-divergence.

Offline RL policies from different algorithms are evaluated in three criteria, policy performance $\rho(\pi)$, violation of the Bellman flow constraint, and violation of the policy correction constraint. Policy performance $\rho(\pi)$ is a return collected by

actually running the offline RL policy on the generated MDP. The violations are quantified using the $L_1$-norm:

$$\text{viol}_{\text{B. F.}} = \sum_s \left|(1 - \gamma)p_0(s) + \gamma(\mathcal{T}_* d_w)(s) - (\mathcal{B}_* d_w)(s)\right|,$$

$$\text{viol}_{\text{P. C.}} = \sum_s \left|\sum_a w(s,a)\pi_D(a|s) - 1\right|.$$

### F.2. Offline RL performance in continuous domain

In this paper, we have described the close relationship between **SemiDICE**, semi-gradient DICE algorithms (**f-DVL**, **ODICE**), and behavior-regularized RL algorithms (**SQL**). We demonstrate that their similarities are also reflected in their practical performance, as they achieve comparable results in continuous domains. We also present the results of **OptiDICE**, which shows limited performance compared to the recent semi-gradient DICE mdthods. We evaluate the performance of the algorithms on D4RL benchmarks (Fu et al., 2020), with fixed hyperparameters as $\alpha = 1$ and $\beta = 0.5$.

| Task | SemiDICE | f-DVL | ODICE | SQL | OptiDICE |
|---|---|---|---|---|---|
| hopper-medium | 66.2 | 63.0 | 86.1 | 74.5 | 46.4 |
| walker2d-medium | 83.4 | 80.0 | 84.9 | 65.3 | 68.1 |
| halfcheetah-medium | 44.7 | 47.7 | 47.4 | 48.7 | 45.8 |
| hopper-medium-replay | 73.8 | 90.7 | 99.9 | 95.5 | 20.0 |
| walker2d-medium-replay | 55.0 | 52.1 | 83.6 | 38.2 | 17.9 |
| halfcheetah-medium-replay | 41.7 | 42.9 | 44.0 | 44.2 | 31.7 |
| hopper-medium-expert | 110.4 | 105.8 | 110.8 | 106.3 | 51.3 |
| walker2d-medium-expert | 109.0 | 110.1 | 110.8 | 110.2 | 104.0 |
| halfcheetah-medium-expert | 93.03 | 89.3 | 93.2 | 39.3 | 59.7 |

## G. Experiment Detail in Continuous Domain

### G.1. Computational Cost

To enable end-to-end training of CORSDICE and baselines, we implemented them in JAX (Bradbury et al., 2018). Utilizing automatic vectorization and just-in-time (JIT) compilation, training CORSDICE for 5 seeds and 5 hyperparameter values over one million gradient update steps took approximately wall-clock time of 5,000 seconds, where the time for training single agent is approximately 200 seconds on a single RTX 3090 GPU. For D-CORSDICE, the pre-training of diffusion-based BC agent took about 2,000 seconds on a single RTX 3090 GPU.

### G.2. Evaluation Protocol

Following DSRL, we evaluate the models with 3 different cost limits. Returns are normalized by the dataset's empirical maximum, and costs by the threshold, where a normalized cost below 1 indicates a safe agent. We increased the training seeds from 3 to 5 and gradient updates from $10^5$ steps to $10^6$ steps to ensure baseline convergence. As in DSRL, we prioritize cost-satifying, safe agents over return-maximizing, unsafe ones. We reported the highest return among safe agents, or if none exist, the return of the least-violating unsafe agents.

For experiments with advanced model, we evaluate algorithms using a single, tighter cost threshold—10 for harder Safety-Gymansium tasks, and 5 for others—averaging results over 3 seeds and 20 episodes.

**Cost Limits**    Following the process of DSRL (Liu et al., 2024), we evaluated the models with 3 different cost limits. For Safety-Gymansium (Ji et al., 2023), cost limits of 20, 40, and 80 were used. For other two environments, Bullet-Gym (Gronauer, 2022) and MetaDrive (Li et al., 2022), we used cost limits of 10, 20 and 40.

**Adjustment of Cost Limit**    While the objective of constrained RL (7) assumes discounted MDP, the actual evaluation is performed with undiscounted sum of costs. This misalignment between the training objective and the evaluation protocol can be remedied by adjusting the cost limit accordingly:

$$C_\gamma = \tilde{C}_{\text{lim}} \cdot \frac{(1 - \gamma^{H+1})}{H}$$

where $H$ is the horizon of the episode. The adjusting coefficient can be derived by assuming the cost function is constant, and this adjustment is also used in DSRL (Liu et al., 2024).

### G.3. Hyperparameters

We used tanh-squashed Gaussian distribution to model the actor, and regular linear layers with ReLU activations (except for the last layer) for critic networks. Following ReBRAC (Tarasov et al., 2024), we utilized Layer Normalization (Ba et al., 2016) in our critic networks. All networks were trained with Adam (Kingma, 2014) optimizers, with the initial learning rate was set to $3e^{-4}$ and scheduled with cosine decay.

In case of D-CORSDICE, we used U-Net (Ronneberger et al., 2015) to train the score model $\epsilon_\theta$ where convolutional layers are replaced with regular linear layers, a common choice in diffusion-based RL (Hansen-Estruch et al., 2023; Mao et al., 2024a). For sampling actions, we used DPM-solver (Lu et al., 2022) and their suggested configuration for sampling from conditional distribution. For training score model, AdamW (Loshchilov & Hutter, 2017) with weight decay of $1e^{-4}$ was used.

We used random search to optimize the hyperparameter for DSRL (Liu et al., 2024) experiment, except for D-CORSDICE where grid search were used to determine the number of actions sampled during inference and the scale of guidance score. Common hyperparameters and their search ranges are summarized in Table 4.

Table 4. Hyperparameters for DSRL (Liu et al., 2024) experiments.

| HYPERPARAMETERS | SAFETYGYM | BULLETGYM | METADRIVE |
|---|---|---|---|
| DISCOUNT FACTOR $\gamma$ | 0.99 | 0.99 | 0.99 |
| BATCH SIZE | 256 | 256 | 256 |
| SCORE BATCH SIZE | 2048 | 2048 | 2048 |
| SOFT UPDATE $\tau$ | $5e^{-4}$ | $5e^{-4}$ | $5e^{-4}$ |
| LEARNING RATES | $3e^{-4}$ | $3e^{-4}$ | $3e^{-4}$ |
| ACTOR HIDDEN DIMS | [256, 256] | [256, 256] | [256, 256] |
| CRITIC HIDDEN DIMS | [256, 256] | [256, 256] | [256, 256] |
| VAE HIDDEN DIMS | [400, 400] | [400, 400] | [400, 400] |
| SCORE RESIDUAL BLOCKS | 6 | 6 | 6 |
| SCORE TIME EMBEDDING DIMS | 32 | 32 | 32 |
| SCORE CONDITIONAL EMBEDDING DIMS | 128 | 128 | 128 |
| DICE $\alpha$ RANGES | $[0.001, 1.0] \cup \{2.0, 5.0\} \cup [10.0, 50.0]$ | | |
| GUIDANCE SCALE VALUES | [1.0, 2.0, 4.0] | [1.0, 2.0, 4.0] | [1.0, 2.0, 4.0] |
| NUMBER OF INFERENCE ACTIONS VALUES | [1, 32, 64, 128] | [1, 32, 64, 128] | [1, 32, 64, 128] |

## H. Additional Experiment Results

### H.1. Ablation Studies on the Cost Sensitivity

Effective offline constrained reinforcement learning (RL) should exhibit *predictable* performance across different cost limits. Ideally, as the cost limit decreases, the algorithm should utilize fewer costs while maintaining minimal declines in return. Figure 2 summarizes these results. Our method, CORSDICE, demonstrated this predictable behavior, effectively using lower costs as the cost limit decreased. In contrast, other baseline methods exhibited inconsistent behavior, sometimes even incurring higher costs despite stricter cost constraints.

### H.2. Additional Experiments on Off-Policy Evaluation

We performed additional experiment to test the off-policy evaluation performance of our extraction method for different regularization strength $\alpha \in \{1, 2, 5\}$. The results are summarized in Figure 3. Our extraction method consistently reduces the RMSE, regardless to the choice of $\alpha$.

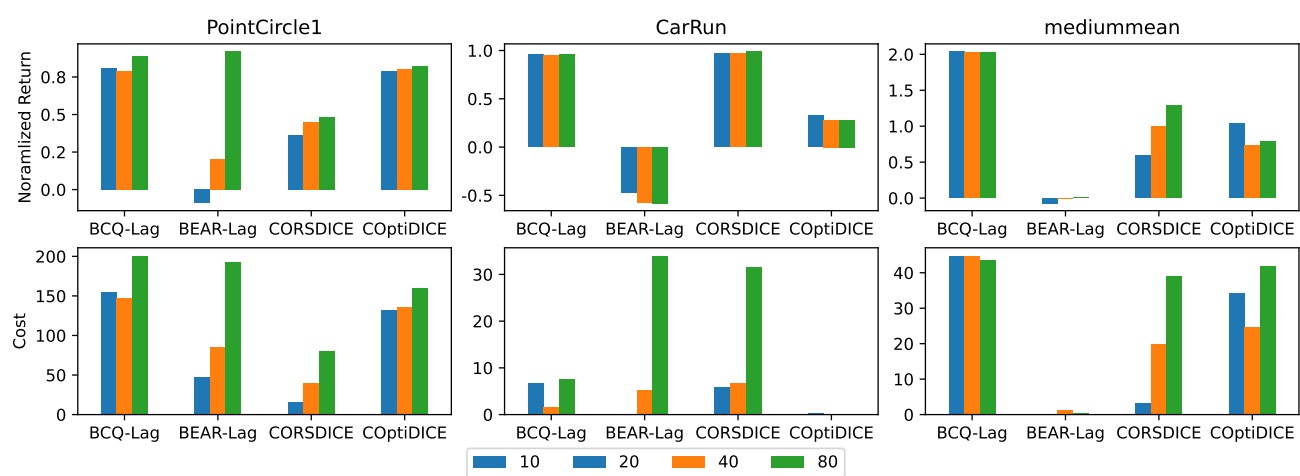

*Figure 2.* Ablation on the sensitivity of constrained RL algorithms on 3 different cost limits. While CORSDICE shows consistent and predictable behaviors, other baselines were inconsistent.

### H.3. Learning Curves on Partial Environments

We included the learning curves of CORSDICE, including some of the baselines, to compare the convergence speed and stability on four environments, summarized in Figure 4.

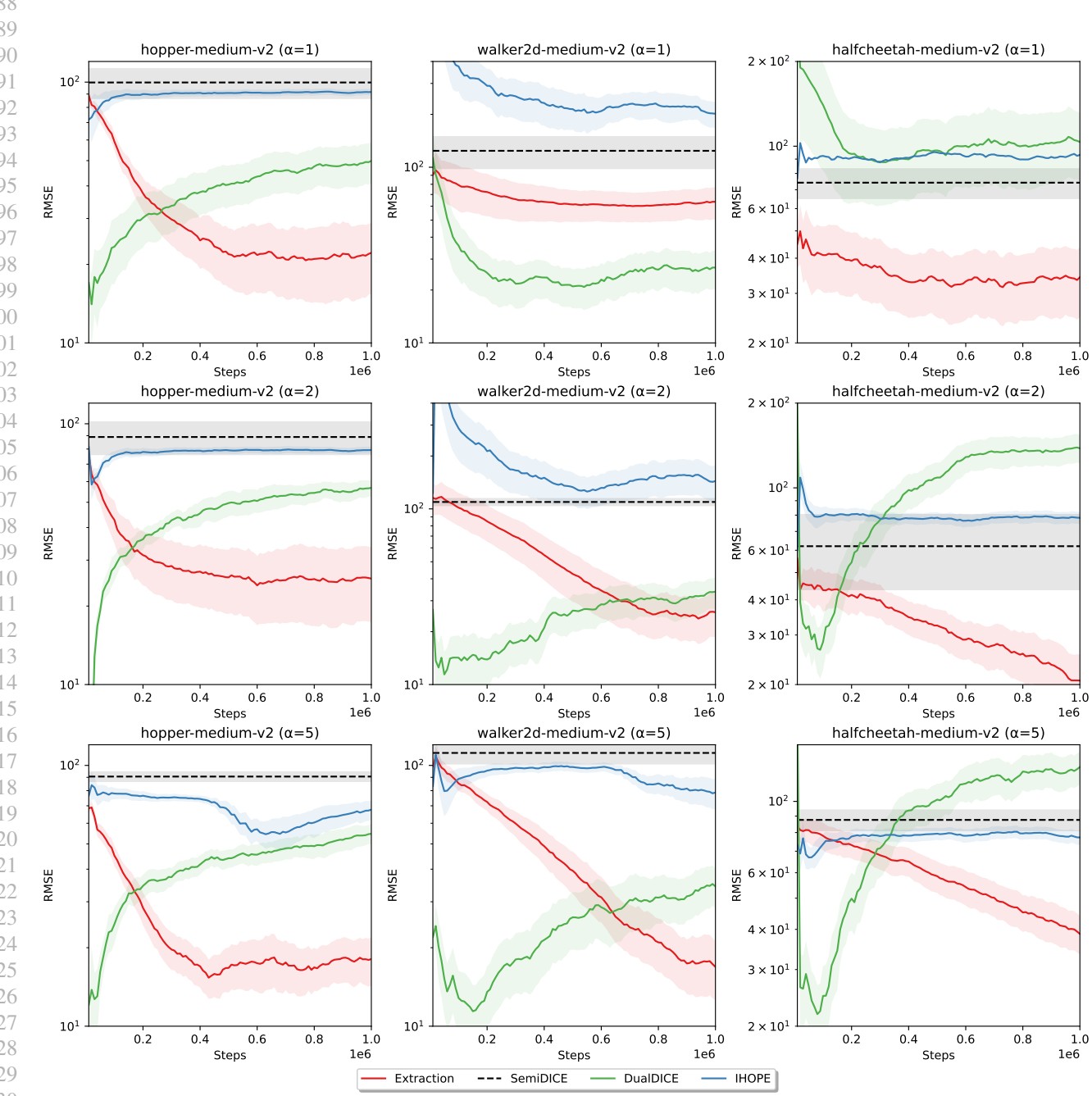

*Figure 3.* Root mean squared error (RMSE) of off-policy evaluation of SemiDICE policy, with different hyperparameters $\alpha$. Our extraction method is robust to the choice of $\alpha$,

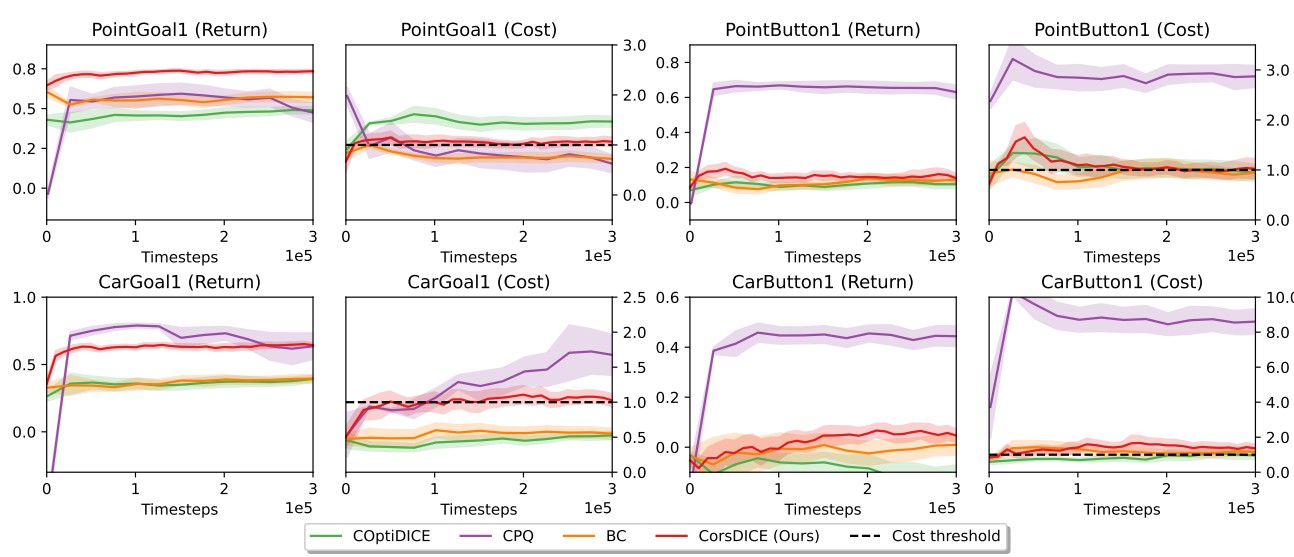

*Figure 4.* Early learning curves of CORSDICE and baselines on four Safety Gymnasium (Liu et al., 2024) tasks, cost limits set to 40. Our method, CORSDICE, shows fast and stable convergence.

