# OpenReview forum: "Semi-gradient DICE for Offline Constrained Reinforcement Learning"
_ICML.cc/2025/Conference — Submitted to ICML 2025_

### Official Review · Reviewer_VB9D · 2025-03-09

**Overall Recommendation:** 4

**Summary:**

This paper investigates the limitations of SemiDICE in offline constrained reinforcement learning, revealing that it outputs policy corrections rather than stationary distribution corrections. This fundamental flaw significantly impairs SemiDICE's effectiveness in off-policy evaluation (OPE). Based on these findings, CORSDICE is proposed to facilitates OPE by computing state stationary distribution corrections.

## update after rebuttal

Thank the authors for offering the empirical evidence. I have no further questions and will keep my score.

**Claims And Evidence:**

While the majority of the claims presented in this paper are substantiated by clear and compelling evidence, one particular assertion in Section 4 (Page 4) requires further clarification. The authors state that "there often exists a state s such that $d_{\pi^{\star}}(s, a) = 0$ for all a in OptiDICE." However, upon careful examination, I was unable to locate any empirical evidence or theoretical justification supporting this specific claim within the manuscript.

**Essential References Not Discussed:**

No

**Experimental Designs Or Analyses:**

The experimental design demonstrates both comprehensiveness and persuasiveness, reflecting the authors' substantial efforts in empirically validating CORSDICE. The experimental evaluation comprises three comprehensive components. The initial component systematically examines algorithmic characteristics, providing empirical validation for the limitations of existing DICE methods as discussed in Section 4. The subsequent component demonstrates CORSDICE's superior capability in achieving precise OPE. The third component evaluates CORSDICE's performance among benchmarks in offline constrained reinforcement learning.

**Methods And Evaluation Criteria:**

It makes sense for problem.

**Other Comments Or Suggestions:**

No

**Other Strengths And Weaknesses:**

No

**Questions For Authors:**

The question I asked in the claims and Evidence part.

**Relation To Broader Scientific Literature:**

The paper is well-grounded in the broader literature.

**Theoretical Claims:**

I’ve checked the correctness of theorem proof mentioned in the main paper.

---

> ### Author Rebuttal · Authors · 2025-04-01
>
> We appreciate the reviewer’s comment and provide empirical evidence in Section 4 to better substantiate our work.
>
> ## Q1. Empirical evidence on the claim from Section 4
>
> We demonstrate that OptiDICE can yield a state $s$ where $d_{\pi^*}(s,a)=0\;\forall a$. The figure provided in https://imgur.com/a/7zh2zlg is based on the Four Rooms domain from Figure 1 of OptiDICE [1], using 1000 trajectories collected under a behavior policy with 0.7 optimality. In this setup, red arrows indicate the optimal policy, and the heatmap shows the state visitation frequency under the optimal policy, $\sum_{a}d_{\pi^*}(s,a)$. While all states are visited by the behavior policy, some states have no arrows, indicating they are not visited by the optimal policy ($\sum_{a}d_{\pi^*}(s,a)\approx 0$). Consequently, the optimal policy in those states cannot be recovered by computing $d_{\pi^*}(s,a)/\sum_{a}d_{\pi^*}(s,a)$. We will include this demonstration in the final version.
>
> [1] Optidice: Offline policy optimization via stationary distribution correction estimation. ICML, 2021.

---

> > ### Comment · Reviewer_VB9D · 2025-04-04
> >
> > Thank you for offering the empirical evidence. I have no further questions and will keep my score.

---

### Official Review · Reviewer_UUBt · 2025-03-13

**Overall Recommendation:** 4

**Summary:**

The paper developed a new offline RL algorithm that applies semi-gradient DICE, addressing the challenge of constraint violation when applying semi-gradient DICE in the context of constrained RL. The paper provides theoretical analysis on the characteristics of the correction term (i.e., the ratio of the stationary distribution w.r.t the learning policy to that w.r.t the dataset policy) that lead to the violation of the Bellman equation (thus the constraint violation). The paper then proposes a stationary distribution correction idea to address the issue.

Experiments on several benchmarks show the advantages of the proposed algorithm CORDICE in comparison with other offline RL algorithms.

**Claims And Evidence:**

The advantages of the proposed algorithm CORDICE are supported by both theoretical and experimental results. In particular, the paper provides substantial empirical analysis on different aspects (i.e.,violations of Bellman and policy correction constraints) and performance of CORDICE comparing with other baselines.

**Essential References Not Discussed:**

I am not aware of missing essential references.

**Experimental Designs Or Analyses:**

Experimental result analysis in the paper is extensive, examining the performance and characteristics of the proposed algorithm in various settings using both D4RL and DSRL datasets.

**Methods And Evaluation Criteria:**

The semi-DICE approach is well-suited for offline RL. Benchmarks such as the D4RL datasets or the DSRL datasets are commonly used in literature.

**Other Comments Or Suggestions:**

None.

**Other Strengths And Weaknesses:**

A key strength of the paper is a thorough theoretical analysis on properties of semi-gradient DICE in constrained RL, including the analysis on the violation of the stationary distribution and the connection with behavior-regularized offline RL. The proposed idea of extracting state stationary distribution to fix the cost violation is novel and well justified.

**Questions For Authors:**

I don't have questions for the authors.

**Relation To Broader Scientific Literature:**

The paper develops a new offline RL algorithm, utilizing DICE --- a well-known framework used in RL literature. There is a long line of research that applies DICE to different settings, ranging from single-agent to multi-agent, from unconstrained to constrained RL. This work extends semi-gradient DICE to the context of constrained RL, offering insights that could be valuable to the RL community.

**Theoretical Claims:**

No, I didn't check the correctness of any proofs in the appendix.

---

> ### Author Rebuttal · Authors · 2025-04-01
>
> We appreciate the reviewer’s acknowledgment of our contribution: extending semi-gradient DICE to constrained offline RL, grounded in theoretical analysis of its optimal solution and stationary distribution correction approach to address the issue of Bellman flow constraint violation.

---

### Official Review · Reviewer_G53g · 2025-03-14

**Overall Recommendation:** 2

**Summary:**

This paper proposes a DICE-based algorithm for offline constrained RL. The proposed method can be seen as a SemiDICE version of COptiDICE, with some extra designs. The paper is generally well-written, but I also feel there is some overclaiming of contributions and a lack of adequate acknowledgment of existing works. The final safety control performance is not very impressive compared to some recent SOTA-safe offline RL algorithms.

**Claims And Evidence:**

- The paper claims to identify the root cause of the limitation when using the SemiDICE framework to perform OPE. The claim is generally supported by the theoretical analysis, although I think the description overstates some points.

- The paper claims that it achieves state-of-the-art (SOTA) performance on DSRL benchmarks, however, by inspecting Table 2 and 3, many results for CORSDICE have somewhat high costs, and many are close to the cost threshold 1. This is not a desirable property for a safe policy.

**Essential References Not Discussed:**

See my comment in the Theoretical Claims section.

**Experimental Designs Or Analyses:**

- The experiment design is generally reasonable, however, I feel the safety control performance of the proposed method is not very impressive. Although the proposed method can control cost value below the threshold, it often gets high cost values (close to the threshold), especially in Table 2.
- Why many tasks in Table 2 are not tested and reported in Table 3?

**Methods And Evaluation Criteria:**

The method generally makes sense. The evaluation on the DSRL benchmark is also reasonable.

**Other Comments Or Suggestions:**

- From G.3, it seems the paper conducted heavy hyperparameter tuning to get good results. This is not encouraged in offline RL research. As most offline RL methods are designed to solve real-world problems that have restricted online system interaction during the training stage. Hence we do not have much opportunity for exhaustive hyperparameter tuning, and bad policy can cause severe consequences on real systems. It is desirable to report results based on fixed hyperparameters or only a small set of hyperparameters. Using random search and hyperparameter optimization to get nicer results is a bad practice in offline RL research.

**Other Strengths And Weaknesses:**

**Strengths:**
- The paper is generally well-written and easy-to-read. The key ideas are clearly conveyed and discussed.

**Weaknesses:**
- The paper actually borrowed lots of methodological designs and insights from existing papers, but does not adequately acknowledge them. For example, the problem framework is from CoptiDICE [1]; the insights and techniques of SemiDICE are from DualRL [2] and ODICE [3]; the insights between SemiDICE and behavior-regularized offline RL is from ODICE [3]; and the trick for bias reduction is from PORelDICE [4]. None of these are adequately stated in the paper.

[1] COptiDICE: Offline Constrained Reinforcement Learning via Stationary Distribution Correction Estimation

[2] Dual RL: Unification and New Methods for Reinforcement and Imitation Learning. ICLR 2024

[3] Revealing the mysteries of distribution correction estimation. ICLR 2024.

[4] Relaxed Stationary Distribution Correction Estimation for Improved Offine Policy Optimization. AAAI 2024.

**Questions For Authors:**

N/A

**Relation To Broader Scientific Literature:**

Safe offline RL has broad applications in robotics, autonomous driving, and industry control.

**Theoretical Claims:**

I think there are some overclaiming and lack of acknowledgments of existing works in the paper.
- For Proposition 4.1, it is well-known that SemiDICE methods may not satisfy the Bellman flow constraint, since it replaces $(1-\gamma)E_{s_0\sim p_0}[v(s_0)]$ with $E_{(s,a)\sim d_D}[v(s_0)]$  and use the parameter $\alpha$, which causes the Bellman flow property to break apart. This is obvious and more or less mentioned in existing works, hence I do not think it is that new.
- For the discussion in Section 4 about the connections to behavior-regularized offline RL, this actually has been thoroughly discussed in the ODICE paper [1] by Mao, et al. (2024). Actually, they derive their methods by first noticing the relationship between SemiDICE and behavior-regularized offline RL. This is not adequately acknowledged in the paper.
- The introduction of another function approximator $A(s)$ for bias reduction is identical to the trick used in PORelDICE [2]. Again, this is never mentioned nor discussed in the description of the methodology, even if the authors actually cited the paper in the preliminary section.

[1] Revealing the mysteries of distribution correction estimation. ICLR 2024.

[2] Relaxed Stationary Distribution Correction Estimation for Improved Offine Policy Optimization. AAAI 2024.

---

> ### Author Rebuttal · Authors · 2025-04-01
>
> We thank the reviewer for the thorough and constructive comments. We hope we can address your concerns below.
>
> ## Q1. Violation of Bellman flow (BF) constraint and the originality of Proposition 4.1
>
> We respectfully disagree with the reviewer’s claim that replacing the term $(1-\gamma)p_0$​ with the dataset distribution $d_D(s)$ is a well-known violation of the BF constraint. To clarify, there are two distinct scenarios for this replacement, under the full-gradient DICE method:
>
> - Direct substitution: While replacing $(1-\gamma)p_0$ with $d_D$ does violate the BF constraint, the resulting d becomes a **scaled (due to removal of $1-\gamma$ factor)** stationary distribution under a modified MDP with initial state distribution $d_D$ **(due to $p_0\to d_D$)**. The constant scaling can be easily corrected during the policy extraction.
> - Assumption-based substitution (as used in SemiDICE, appx. B, Eq. (31-32)): Here, we assume $d_D$ satisfies the stationary distribution condition. This method does __not__ violate the Bellman flow constraint.
>
> Thus, we argue that the fundamental cause of BF violation is the use of semi-gradient update—not the replacement of the initial state distribution. To our knowledge, this is the first time this specific cause has been clearly identified.
>
> Furthermore, Proposition 4.1 does more than pinpoint the violation's cause. It shows that SemiDICE outputs the policy ratio, which is essential for applying our stationary distribution extraction technique. Without Proposition 4.1, this insight would not be possible. We believe this adds a meaningful contribution to the understanding of semi-gradient DICE methods.
>
> Finally, α only affects the conservatism strength of DICE method—it has no impact on the BF property.
>
> ## Q2. Lack of acknowledgement [1, 2] for discovering the connection
>
> Although we cited [1, 2] and briefly discussed them in appx. C, we will revise the manuscript to better acknowledge their discovery. While earlier studies implied the connection through similar loss functions, we make it more explicit by showing that a behavior-regularized MDP with a general f-divergence can be approximated by its corresponding SemiDICE. This insight is central to our work, as it justifies the SemiDICE policy ratio and supports its constrained extension—the main contribution of our paper.
>
> ## Q3. Lack of acknowledgement [3] for bias reduction technique
>
> While our bias reduction technique was inspired by PORelDICE [3], we approximate a different expectation. PORelDICE estimates $U(s,a)\approx r(s,a) +\gamma\sum_{s'}T(s'|s,a)\nu(s')$, while we approximate $A(s)\approx\sum_a\pi(a|s)(\gamma\sum_{s'}T(s'|s,a)\mu(s')-\mu(s))$. Consequently, PORelDICE focuses on reducing transition bias to improve offline RL, whereas our approach targets bias from both the transition dynamics and the policy to improve OPE. We will acknowledge [3] appropriately and clarify this distinction in the final version.
>
> ## Q4. High cost close to limit
>
> We respectfully clarify that obtaining cost values near the constraint threshold is not a flaw but a desirable behavior in our average-constrained optimization setting (CMDP, Eq. (7)). Our objective explicitly encourages using as much of the cost budget as needed to improve the return, so long as the expected constraint is satisfied. This is a principled difference from hard or probability-constrained formulations, where any thresholds violation is unacceptable. Recent works (e.g. [4], [5]) have studied these stricter formulas; our method is complementary and could be extended to them in the future. We will update the manuscript to make this distinction clearer.
>
> ## Q5. Hyperparameter tuning
>
> While G.3 lists all potential hyperparameters, we only tuned one—α—for CORSDICE. For D-CORSDICE, we tuned two additional hyperparameters (guidance scale values, the number of inference action values), which were directly adopted from D-DICE [6]. For inference samples, we used a smaller search space than in D-DICE (Table 4 of [6]). The original paper did not specify a search space for guidance scale, so we referred to the values in their official implementation. Importantly, all methods, including baselines, were tuned with the same hyperparameter search budget and procedure to ensure fair comparison. We will clarify these points in the revised manuscript.
>
> [1] Dual RL: Unification and New Methods for Reinforcement and Imitation Learning. ICLR 2024
>
> [2] ODICE: Revealing the Mystery of Distribution Correction Estimation via Orthogonal-gradient Update. ICLR 2024
>
> [3] Relaxed stationary distribution correction estimation for improved offline policy optimization. AAAI 2024
>
> [4] Safe Offline Reinforcement Learning with Feasibility-Guided Diffusion Model. ICLR 2024
>
> [5] Quantile Constrained Reinforcement Learning: A Reinforcement Learning Framework Constraining Outage Probability. NeurIPS 2022
>
> [6] Diffusion-DICE: In-Sample Diffusion Guidance for Offline Reinforcement Learning. NeurIPS 2025

---

### Decision · Program_Chairs · 2025-05-01

**Decision:**

Reject

**Comment:**

The reviewers recognize the thoroughness of both the theoretical analysis and the experimental campaign in diagnosing and proposing solutions to subtle issues of DICE approaches.
At the same time, since several recent papers address related problems and present related solutions, some further work should be put into the presentation in order to clearly position the contribution within the recent DICE literature. In particular, regarding insights from [1] and [2] (e.g., the connection between SemiDICE and behavior-regularized offline RL) and algorithmic solutions from [3], [4]. Besides crediting these previous works, the overall presentation should be adjusted to provide a clear picture to the reader.

[1] Dual RL: Unification and New Methods for Reinforcement and Imitation Learning. ICLR 2024
[2] ODICE: Revealing the Mystery of Distribution Correction Estimation via Orthogonal-gradient Update. ICLR 2024
[3] Relaxed stationary distribution correction estimation for improved offline policy optimization. AAAI 2024
[4] COptiDICE: Offline Constrained Reinforcement Learning via Stationary Distribution Correction Estimation